# A novel environment-evoked transcriptional signature predicts reactivity in single dentate granule neurons

Baptiste N. Jaeger [1,2], Sara B. Linker[1], Sarah L. Parylak[1], Jerika J. Barron [1], Iryna S. Gallina[1], Christian D. Saavedra[1], Conor Fitzpatrick[1], Christina K. Lim[1], Simon T. Schafer[1], Benjamin Lacar[1], Sebastian Jessberger[2] & Fred H. Gage [1]

Activity-induced remodeling of neuronal circuits is critical for memory formation. This process relies in part on transcription, but neither the rate of activity nor baseline transcription is equal across neuronal cell types. In this study, we isolated mouse hippocampal populations with different activity levels and used single nucleus RNA-seq to compare their transcriptional responses to activation. One hour after novel environment exposure, sparsely active dentate granule (DG) neurons had a much stronger transcriptional response compared to more highly active CA1 pyramidal cells and vasoactive intestinal polypeptide (VIP) interneurons. Activity continued to impact transcription in DG neurons up to 5 h, with increased heterogeneity. By re-exposing the mice to the same environment, we identified a unique transcriptional signature that selects DG neurons for reactivation upon re-exposure to the same environment. These results link transcriptional heterogeneity to functional heterogeneity and identify a transcriptional correlate of memory encoding in individual DG neurons.

[1] The Salk Institute for Biological Studies, La Jolla, CA 92037-1002, USA. [2] Laboratory of Neural Plasticity, Faculty of Medicine and Science, Brain Research Institute, University of Zurich, 8057 Zurich, Switzerland. These authors contributed equally: Baptiste N. Jaeger, Sara B. Linker, Sarah L. Parylak Correspondence and requests for materials should be addressed to F.H.G. (email: gage@salk.edu)

Neuronal circuits are constantly remodeled in response to activity. Activity-induced plasticity allows the brain to adjust to changing conditions, as in development, adaptation after injury, or learning. Within a given circuit, the molecular consequences of activity may differ by cell type.

One well-characterized circuit lies in the hippocampus, a structure critical for learning and memory. Entorhinal cortex provides input to the DG, which sends outputs to CA3, which in turn transmits information to CA1[1,2]. The principal cell types in these regions vary in the number of cells activated both at baseline and when stimulated by exposure to novelty. Compared to pyramidal cells of CA3 and CA1, dentate granule (DG) neurons have particularly sparse activity. DG neurons spike less frequently[3–6], express immediate early genes (IEGs) in fewer cells[7–9] and have fewer calcium transients[10].

The consequences of activity also differ across hippocampal subfields. Long-term potentiation (LTP) shows considerable variation within the hippocampus. LTP of the Schaffer collaterals from CA3 to CA1 is NMDA-receptor dependent and expressed postsynaptically[11,12]. In contrast, mossy fiber LTP from the DG to CA3 is NMDA-receptor independent and expressed presynaptically[13,14]. Non-synaptic forms of plasticity also show regional variation. In CA1, activation by either electrical stimulation or learning produces a temporary increase in cell excitability[15–18]. The same paradigms fail to change the excitability of DG neurons[19], although recent work has demonstrated that artificially enhancing DG neuron excitability influences which cells activate[20].

Cell-type specificity of activity levels and plasticity mechanisms suggests that later molecular processes may also differ across populations, including long-term changes underlying memory formation. Memory formation is a process, not an instantaneous event[21]. Activity-induced changes, including RNA and protein synthesis, are initiated during a salient experience, and perturbing these changes over subsequent minutes, hours, and days influences the stability and robustness of the memory of that experience[22]. The molecular actors enlisted during memory formation are likely tailored to the needs of each cell. A fraction of the population active during memory encoding is later reactivated during memory recall. Optogenetic stimulation of these engram cells activates memory recall whereas silencing them impairs memory[23,24]. Although these features of engram cells are common across a variety of brain regions, the probability of reactivation is region specific. Such cell type-specific properties rely, at least partially, on differences in transcription[25,26].

Until recently, molecular mediators of memory formation were studied in bulk populations. Activity was induced physiologically (e.g., electrical stimulation, seizure) or behaviorally (e.g., exposure to a new environment, contextual fear learning), and a search was conducted for genes or proteins with modified population-wide expression. This approach identified numerous IEGs. However, beyond canonical examples such as c-fos (FOS), IEGs from different regions, times post activation, or stimulation paradigms have surprisingly little overlap[27–30]. Now, sequencing technology permits transcriptome analysis of single neurons[31,32], and we recently developed a method for sequencing single nuclei that preserves activity-dependent gene expression[33]. Our technique allows activated and non-activated nuclei to be isolated based on IEG protein expression that would be difficult to detect using whole cell methods[34]. This technique dramatically improves our ability to understand the source of variability in activity-induced processes.

Here, we sought to understand activity-dependent transcriptional changes in single nuclei that lay the groundwork for memory formation in the hippocampus (Supplementary Fig. 1). One hour after exposing mice to a novel environment (NE), we isolated activated (FOS+) and non-activated (FOS−) neurons from multiple hippocampal populations. We then performed single nucleus RNA-seq (snRNA-seq) using an unbiased deep sequencing approach that enabled high-resolution quantification of high-, mid-, and low-expressing genes. Next, we assessed the late transcriptional response in activated DG neurons 4–5 h after NE exposure by taking advantage of the sustained expression of activity-regulated, cytoskeleton-associated protein (ARC) in DG neurons. Finally, we sought to relate heterogeneity in the late activity-induced transcriptional response to the selection of putative engram cells among the original activated population. Four hours after exposing mice to an initial NE, we exposed them to either the same or a different NE and collected activated nuclei 1 h later. By comparing the transcriptome of DG neurons activated in the two contexts, we identified a unique transcriptional signature that selects DG neurons for reactivation. A computational model built on this signature allowed us to predict whether an individual neuron was likely to become an engram cell and reactivate upon re-exposure to the same environment.

## Results

**Exposure to a NE triggers activation across the hippocampus.** We designed an antibody panel to capture hippocampal nuclei from DG, CA, and interneurons. Anti-NEUN distinguishes neurons from glia, anti-PROX1 labels DG neurons[35], and anti-CTIP2 labels both CA1 and DG[36]. This panel allowed for the putative discrimination of DG (NEUN+PROX1+CTIP2+) and CA1 neurons (NEUN+PROX1−CTIP2+) and a population of NEUN+PROX1−CTIP2− neurons (Negs) containing CA3, CA2, and interneurons (Fig. 1a). Vasoactive intestinal polypeptide (VIP) interneurons express NEUN and PROX1 but lack CTIP2 (Fig. 1b), enabling isolation of a defined population of GABAergic nuclei[37,38]. We next dissected the hippocampus of home cage (HC) mice, isolated nuclei by Dounce homogenization, and performed antibody staining. Using flow cytometry, we could identify all four neuronal populations (DG, CA1, VIP, and Negs) (Fig 1c, d).

To identify active cells we stained for the IEG FOS[39]. In HC mice the percentage of FOS+ neurons revealed the low baseline activity of DG neurons (0.3% SD ± 0.09) compared to CA1 and the Negs (2.2% SD ± 1.9 and 1.8% SD ± 1.6, respectively), whereas a larger fraction of VIP interneurons was active (4.4% SD ± 1.7). To assess activation to a naturalistic stimulus, mice were placed in a NE for 15 min and then returned to their HC until sacrifice 1 h later (Fig. 1e). FOS was induced across the hippocampus, but the relative activity of each population was preserved. DG neuron activation was particularly sparse relative to all other populations (1.6% SD ± 0.2). CA1 neurons and the Negs showed moderate levels of activation (15.1% SD ± 3.9 and 12.3% SD ± 4.2, respectively), whereas VIP interneurons responded the most (32.3% SD ± 3.5) (Fig 1f, g). These populations thus spanned a range from sparsely to frequently active following the same behavioral stimulus: DG < CA1 < VIP (Fig. 1g).

**Active DG neurons exhibit a dramatic shift in transcription.** To dissect cell type-specific transcriptional responses to activity, we isolated FOS+ and FOS− single nuclei from the populations in Fig. 1d and prepared RNA following the SmartSeq2 protocol[32,40]. Nuclei were excluded as outliers based on total aligned reads and total gene count, or extreme-outliers, based on clustering (Supplementary Fig 2a–b). An average of 1.17 million reads were aligned per nucleus, with an average of 5637 genes detected above a $\log_2(\text{TPM}+1)$ (TPM) = 1. We identified both GABAergic neurons (*Gad2+*), including VIP, Parvalbumin, and Ivy interneurons[41], and glutamatergic neurons, including DG, CA1, CA3,

and subiculum (Supplementary Fig 2c, Supplementary Data 2). Biological replicates from multiple mice clustered based on cell type, indicating that batch effects were minimal drivers of clustering (Supplementary Figs 2b and d).

T-distributed stochastic neighbor embedding (t-SNE) analysis of DG, CA1, and VIP neurons after NE exposure revealed a striking separation between DG FOS+ and FOS− neurons, despite the close association within FOS+ or FOS− groups separately. Conversely, FOS+ and FOS− CA1 neurons clustered closely, and the positions of FOS+ and FOS−VIP interneurons directly overlapped (Fig. 2a). Due to the intrinsic difference in baseline activity between cell types, some populations were more likely to contain nuclei that were active in the HC independent of NE exposure. We noted a subset of CA1 neurons that was negative for FOS protein but expressed the canonical IEG *Arc*. To compare neurons that were activated during the NE (FOS+) to an inactive baseline population, nuclei stained as FOS− but exhibiting high expression of *Arc mRNA* (TPM > 2.5) were excluded from downstream analysis (11 CA1, 0 DG, and 0 VIP).

We identified 749, 39, and 3 differentially expressed genes (DEGs) between FOS+ and FOS− nuclei (ROTS $p_{adj} < 0.05$) in DG, CA1, and VIP neurons, respectively (Fig. 2b, Supplementary Data 3). These values represent approximately 13% (DG), 0.5% (CA1), and 0.05% (VIP) of the average number of genes detected in each population (TPM > 1), indicating a striking quantitative difference in the transcriptional response to activity in DG neurons compared to CA1 or VIP. Varying DEG counts were not driven by differences in sample size between groups. When subsampling each cell type to 20 nuclei ($N_{,FOS+} = 10$, $N_{,FOS-} = 10$), DEG counts remained significantly associated with cell type (ANOVA $p < 4.01e−07$), with a 25-fold higher DEG count in DG compared to CA1 (DG: 99.8 ± 27.0; CA1: 4.0 ± 1.7; VIP: 1.8 ± 0.8). To determine whether both FOS+ and FOS− neurons were transcriptionally modified by NE exposure, transcription levels were compared to FOS− neurons from HC animals. The majority of activity-dependent genes were modified only within FOS+ neurons (Fig. 2c). The early cellular response in FOS+ neurons resulted in both induction and repression of gene expression.

In HC mice, a subset of DG neurons displayed FOS at a detectable but lower level (FOS low) compared to that observed following NE exposure (Supplementary Fig. 3a). We examined their activity-induced expression in comparison to FOS+ and FOS− DG neurons using Monocle. (Supplementary Fig. 3b). Similar to the FOS protein stain, FOS low DG neurons from both HC and 1-h animals were intermediate between FOS− and FOS+. Furthermore, while top DEGs such as *Arc* and *Inhba* were increased in FOS low cells, expression was lower than in FOS+ and more variable (Supplementary Fig. 3c).

We next considered the overlap of activity-induced transcription between cell types. As expected, all cell types exhibited higher *Fos* RNA in the FOS+ nuclei (DG: ROTS $p_{raw} < 3.45e−05$; CA1 ROTS $p_{raw} < 1.8e−04$; VIP ROTS $p_{raw} < 6.5e−03$. Figure 2e). DG and CA1 neurons had significantly overlapping DEGs, with canonical IEGs such as *Homer1* and *Nr4a2* increased in both populations (hypergeometric $p < 5.65e−35$; Fig 2d, e). However, these IEGs were not increased in FOS+ VIP interneurons and were not detected within FOS+ GABAergic Pvalb and Ivy interneurons (Fig. 2e and Supplementary Fig. 3d). Despite the presence of FOS protein in all cell types, these results reveal a cell type-specific transcriptional response to activity, with GABAergic VIP interneurons displaying a modest effect compared to glutamatergic CA1 and DG neurons (Supplementary Fig. 3d).

DEGs upregulated in FOS+ DG neurons were assessed for functional enrichment through DAVID bioinformatics using a hypergeometric test. DEGs were enriched for terms such as phosphoproteins ($p_{adj} < 4.40e−20$), positive regulation of transcription ($p_{adj} < 8.5e−05$), neuron projection ($p_{adj} < 3.4e−04$), and learning ($p_{adj} < 2.4e−02$) (Supplementary Data 3). DEGs downregulated in FOS+ DG neurons were enriched for phosphoproteins ($p_{adj} < 6.21e−15$), mitochondrial genes ($p_{adj} < 9.8e−03$) and acetylation genes ($p_{adj} < 3.11e−10$). With a less restrictive threshold of raw $p$-value < 0.05, CA1 and VIP showed overlap in some of these categories, though to a much lower degree (Supplementary Fig 4a and b). These results suggested that DG neurons were robustly altered in response to activity in a way not observed in either CA1 neurons or VIP interneurons.

**FOS+ DG neurons have the highest activity state.** One possible explanation for the unique signature observed in DG neurons is that FOS− neurons from all cell types have the same baseline activity-regulated transcription and then activation shifts DG FOS+ neurons to a level of expression not observed in other cell types. Alternatively, FOS− DG neurons may maintain a uniquely repressed profile of activity-regulated genes and they catch up to the level of CA1 neurons upon stimulation. To facilitate comparison across cell types, we constructed an independent components (IC) plot using Monocle[42,43]. FOS+ and FOS- DG, CA1, and VIP neurons from both HC and NE-exposed mice were plotted using activity-dependent genes identified in any of the three cell types. Relative activity state was then defined as the position along the main trajectory through this graph (Supplementary Fig. 3e). As expected, VIP FOS+ and FOS− neurons were equivalent in activity-related transcription ($t$-test $p = 0.84$). Conversely, both CA1 ($t$-test $p < 6.7e−04$) and DG ($t$-test $p < 2.2e−16$) FOS+ neurons were significantly shifted along the activity axis compared to their corresponding FOS- neurons (Fig. 3a–b). When comparing across cell types, all VIP neurons were lower on the activity axis compared to CA1 and DG neurons. FOS− CA1 neurons displayed a slightly higher transcriptional activity signature than DG FOS− neurons (Fig. 3b), and FOS+ DG neurons displayed the highest activity-induced transcription signature.

Including all cell types (Supplementary Fig. 3e and c) showed that the other interneurons, Ivy and Pvalb, possessed a similar transcriptional signature to VIP (Fig. 3d). CA3 and subiculum expressed activity-dependent genes to a similar degree as CA1, and DG neurons displayed the highest activity-dependent transcriptional signature. Despite their sparse activity, DG neurons have a robust transcriptional response to activity that has the potential to modify future neuronal function (Fig. 3e). Given this transcriptional change and the known importance of the DG in keeping newly encoded memories distinct[44], we chose to investigate the prolonged transcriptional consequences of activity within DG neurons only (Supplementary Fig. 1).

**Activity-induced transcription continues in DG over 5 h.** We hypothesized that activity-induced transcription primes neurons for future reactivity (Supplementary Fig. 1); therefore, it was important to identify gene signatures that developed or persisted after the initial wave of IEGs (e.g., FOS) had subsided. We traced recently activated neurons over time by co-staining for FOS and ARC. Mice were exposed to a NE for 15 min and then returned to their HCs until sacrifice at 1, 5, or 15 h later. Unlike FOS, which is degraded within a few hours, ARC persists for at least 5 h after activation in the DG (Fig. 4a–b, Supplementary Fig 5a–e). One hour after NE exposure, FOS and ARC were colocalized within PROX1+CTIP2+ nuclei (Fig. 4c) and within whole DG cells (Supplementary Fig. 5b). These results confirm previous observations of prolonged ARC expression in the DG[45,46] and validate the use of ARC to track activated DG neurons for up to 5 h after activation.

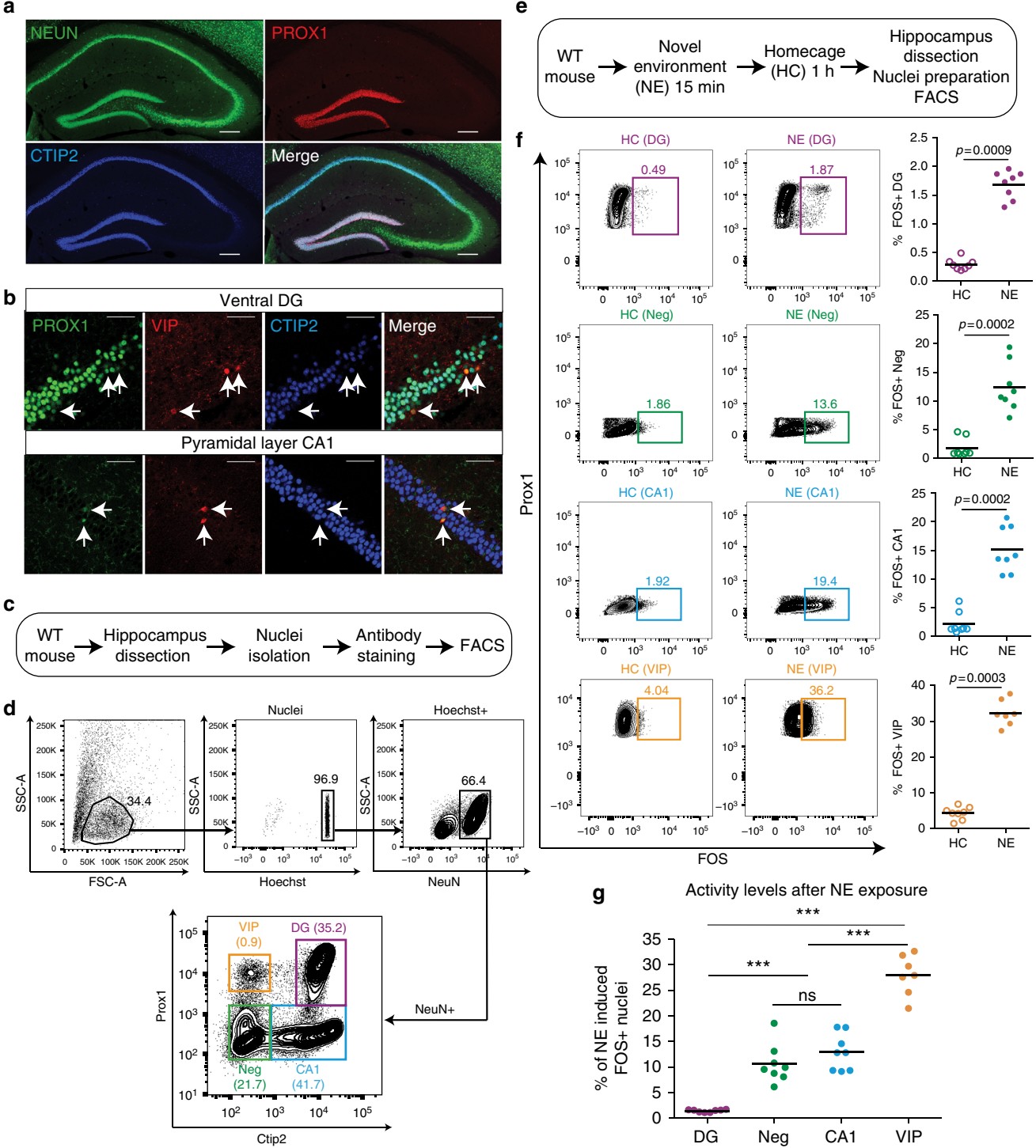

**Fig. 1** Flow cytometry dissection of the hippocampus. **a** Confocal images of NEUN, PROX1, and CTIP2 protein expression in the hippocampus. Scale bars = 200 μm. **b** PROX1+CTIP2−VIP+ interneurons (arrows) are found in the dentate gyrus (DG) and CA1. Scale bars = 50 μM. **c** Workflow for flow cytometry dissection of the hippocampus. **d** Representative FACS plots showing expression of NEUN, PROX1, and CTIP2 in hippocampal nuclei from 7- to 8-week-old mice (n = 8). 65% (SD ± 6%) of isolated nuclei were NEUN+. Among NEUN+ nuclei, 38% (SD ± 4%) were DG neurons (PROX1+CTIP2+), 45% (SD ± 6%) were CA1 (PROX1−CTIP2+), 10% (SD ± 3%) were Negs (PROX1−CTIP2−), and 1% (SD ±0.4%) were VIP interneurons (PROX1+CTIP2−). **e** Workflow for the flow cytometry dissection of the hippocampus following novel environment (NE) exposure. **f** Representative FACS plots showing expression of PROX1 and FOS in each population after home cage (HC) or NE exposure. Percentages of FOS+ nuclei are displayed above gates. At least 200,000 NEUN+PROX1+CTIP2+ nuclei were analyzed per mouse, n = 7–8 mice. P values are indicated for Mann–Whitney test. **g** NE-induced FOS expression in each population. Baseline activation in HC was subtracted from activated levels after NE exposure. n = 7–8 animals. Statistical difference between groups has been determined by one-way ANOVA ($F(3,27)$ = 84.9, $p < 0.0001$) and Tukey's multiple comparisons test (*** = $p < 0.0001$, ns = not significant)

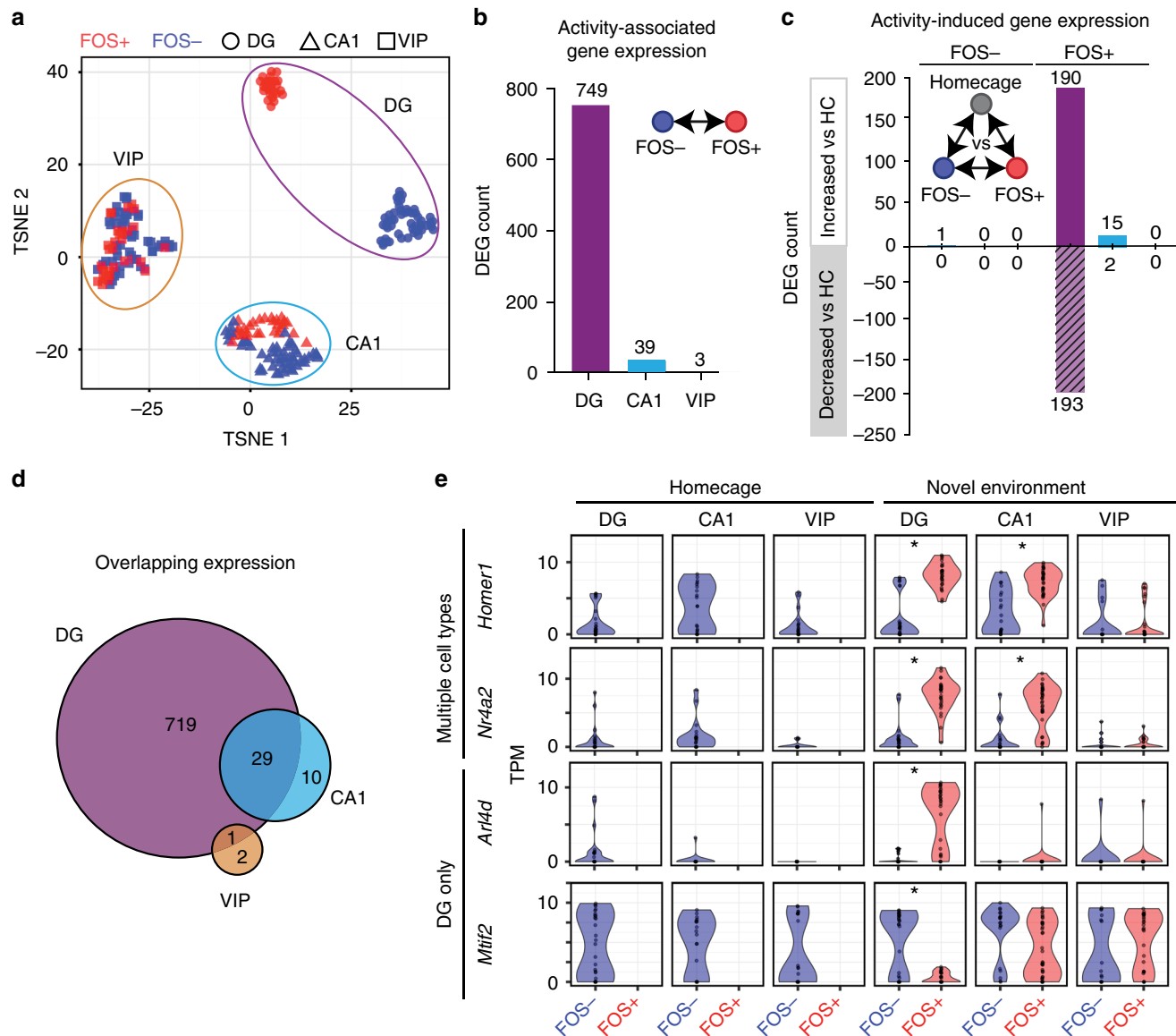

**Fig. 2** Activity-induced gene expression across the hippocampus. **a** T-SNE plot of DG, CA1, and VIP nuclei stained for FOS and isolated from mice exposed to a NE. **b** DEG count between FOS+ and FOS− nuclei for each population in A. **c** DEG count after correcting for HC expression. Bars extending upward and downward represent genes that are increasing or decreasing in the given cell type compared to HC, respectively. **d** Overlap of DEGs across cell types. **e** Representative violin plots of DEGs. Each dot represents a single nucleus. * ROTS $p_{adj} < 0.05$

DG neurons at early (1 h) and late time points (4 and 5 h) were examined using t-SNE (Fig. 4d). The HC FOS- and 1-h FOS− neurons clustered together into a baseline signature, marked by high *Prdm5* expression (Supplementary Fig. 5g). The 1-h FOS+ neurons separated into a distinct cluster denoted as the early signature and expressed IEGs such as *Fosb* (Fig. 4d and Supplementary Fig. 5g). The 4- and 5-h ARC+FOS− nuclei clustered separately and expressed unique genes, including *Sorcs3*, indicating that a separate transcriptional profile continued to develop over time (late signature). To examine the transcriptional dynamics in more detail, we classified the temporal expression patterns of activity-related genes into seven groups (Fig. 4e; Supplementary Data 4). The most prevalent group (group 1; 252 genes) exhibited a canonical IEG dynamic characterized by increased expression at 1 h and a return to baseline by 4 h (e.g., *Fosb*, *Egr1*; Fig. 4f). Group 1 genes were assessed for enrichment using the hypergeometric test and were enriched for terms such as phosphoproteins ($p_{adj} < 8.7e-18$), Ubl conjugation ($p_{adj} < 1.1e$

$-06$), acetylation ($p_{adj} < 2.4e-06$), and regulation of transcription ($p_{adj} < 4.4e-02$), and the promoter regions were enriched for CREB binding sites (HOMER[47] $p_{adj,motif\ enrichment} < 6.3e-03$; STAMP[48] $p_{adj,CREB\ similarity} < 5.00e-12$). A second group of genes (group 5; 107 genes termed sustained) was increased at 1 h and remained elevated at both 4 and 5 h following activity (e.g., *Nptx2*, *Arc*, *Gadd45b*, *Synpo*). Group 5 genes were enriched for phosphoproteins ($p_{adj} < 8.9e-06$), and the terms dendrite ($p_{adj} < 2.00e-04$), synapse ($p_{adj} < 7.50e-03$), and cell projection ($p_{adj} < 2.20-e02$). Additionally, a late signature was present where genes were elevated at both 4 and 5 h following NE but not at 1 h (group 4; 129 genes, e.g., *Sorcs3*, *Dlg2*, *Gabra4*). Group 4 showed the strongest enrichment for genes associated with membrane proteins ($p_{adj} < 1.90e-11$), the postsynaptic membrane ($p_{adj} < 5.00e-05$), and calcium ($p_{adj} < 1.50e-02$). To validate our snRNA-Seq results, we chose two of the late signature genes (*Sorcs3* and *Blnk*) and performed fluorescent in situ hybridization in combination with immunohistochemistry for ARC protein. At

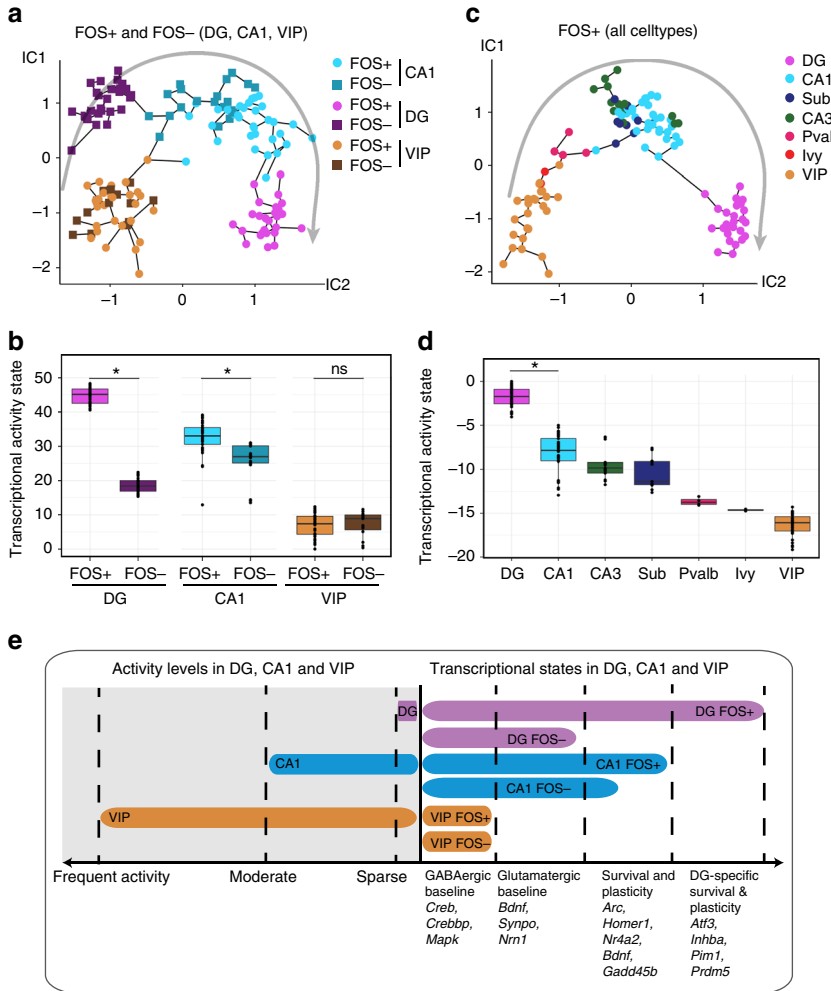

**Fig. 3** DG neurons exhibit a unique transcriptional signature following NE exposure. **a**, **b** Activity-dependent genes were used to construct an independent components (IC) plot using Monocle for DG, CA1, and VIP nuclei after exposure to a NE. **a** Nuclei are colored with respect to cell type. Arrow indicates the direction of increasing *Arc* expression. **b** Position along the main trajectory for each group of neurons. Transcriptional activity state (y-axis) = pseudotime as calculated by Monocle. * = t-test p-value < 0.05, ns = not significant. **c**, **d** Activity-dependent genes were used to construct a second independent Monocle plot for FOS+ neurons from all cell types. **c** Each cell is colored with respect to cell type. Arrow indicates the direction of increasing *Arc* expression. Sub = Subiculum, Pvalb = Parvalbumin interneuron. **d** Position along the main trajectory for each group of FOS+ neurons. * = Student's t-test p-value < 0.05. **e** Conceptual summary of activity-induced expression within cell types with inherently different population activity rates. All cells exhibited expression of baseline genes required for activity-induced expression. DG and CA1 had a slightly elevated baseline of activity-related expression for a small subset of genes. DG and CA1 FOS+ neurons increased expression of known survival and plasticity genes, and DG FOS+ neurons further specifically upregulated an additional set of survival and plasticity-related genes

5 h post NE exposure, we observed substantial colocalization of *Sorcs3* and *Blnk* in ARC+ cells in the DG (Fig. 4g). The transcriptional dynamics induced by brief NE exposure, therefore, continued to develop over several hours, particularly regarding genes that can alter neuronal function, such as synaptic proteins and kinases.

**Gene signatures discriminate reactivation and new activation.** A fundamental advance in understanding memory was the demonstration that memory retrieval preferentially reactivates the same neuronal network that was active during encoding and that these engram cells are necessary and sufficient for the behavioral expression of a memory[23,24]. We sought to identify the transcriptional changes that support reactivation of engram cells by re-exposing the mice to a second context. We expected to observe more neuronal reactivation in mice re-exposed to the same context, given that neurons firing during the first exposure should

receive largely the same input during the second. Conversely, we expected a new set of neurons to activate in a different context, given that the DG has been shown to recruit distinct neuronal ensembles to represent different environments[7]. To confirm the ability of mice to maintain a memory of the first NE at the time of the second exposure, we first quantified their exploratory behavior when exposed to a different context. As we were not able to track the behavior of mice in the exact same environment used for all our previous sorting experiments (NE A), we used two different environments equipped with overhead cameras (A′ and C′). Mice were exposed to a NE (A′) for 15 min and then returned to their HCs. Four hours after the first experience, we re-exposed mice either to the same environment (A′ > A′ group) or to a different environment (A′ > C′) for 15 min (Fig. 5a). Mice in the A′ > A′ group exposed to the same environment 4 h apart showed less exploration during the second exposure, demonstrating their familiarity with the environment. In contrast, mice in the A′ > C′ group showed no habituation of exploratory

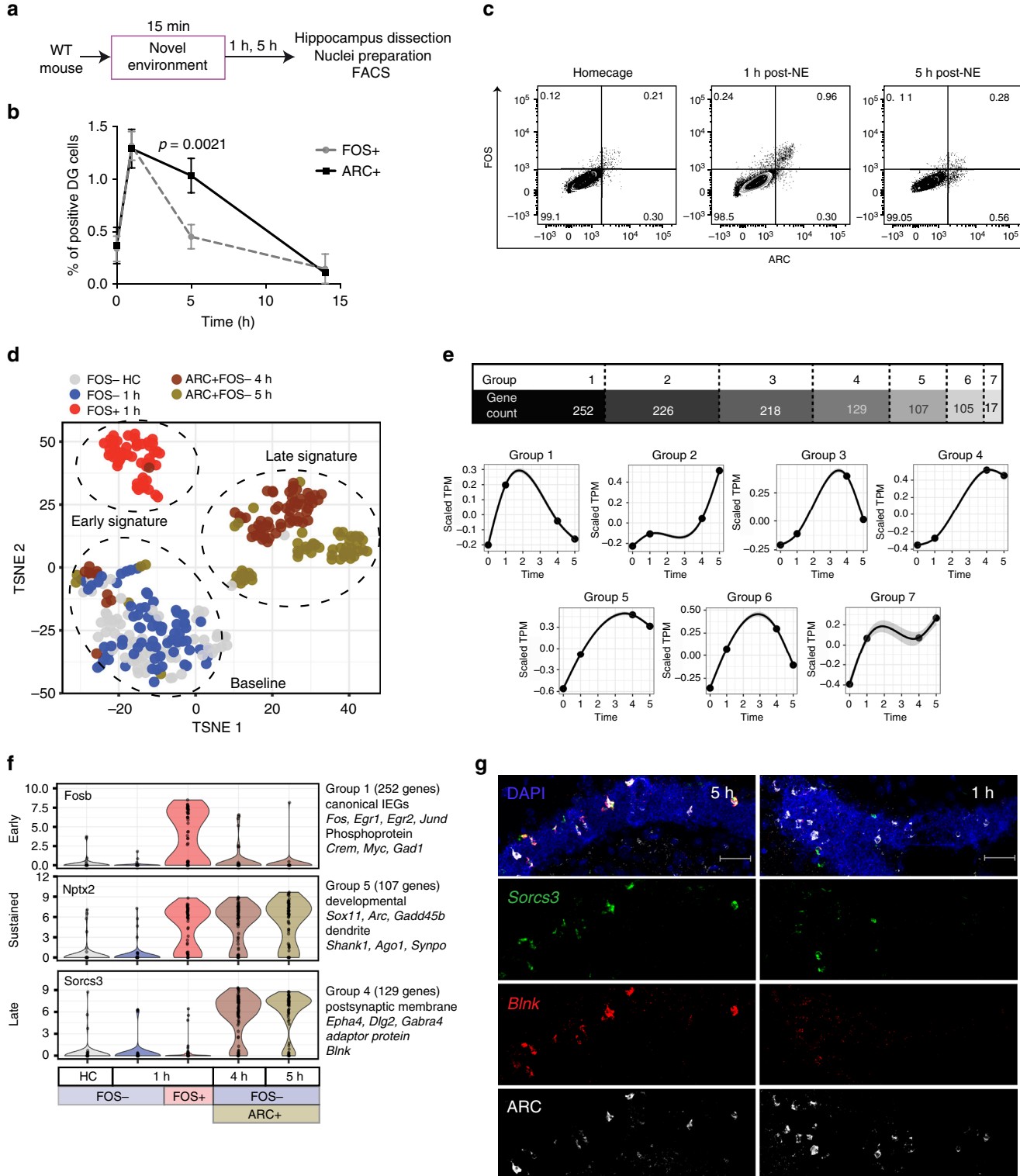

**Fig. 4** A dynamic transcriptional response occurs in DG neurons following NE exposure. **a** Experimental paradigm used for flow cytometry isolation of DG neurons at late time points post NE exposure. **b** Percentage of FOS+ (gray) and ARC+ (black) DG neurons in animals from HC and 1 h, 5 h or 15 h after a 15-min NE exposure ($n = 3$–14 mice per time point, ±S.D), $p$ values are indicated for Mann–Whitney test. At least 150,000 DG nuclei were analyzed per mouse. **c** Representative FACS plots showing FOS and ARC expression in DG neurons (PROX1+CTIP2+ single nuclei) in HC and at 1 h or 5 h post-NE exposure. Percentages of nuclei per gate are indicated. **d** T-SNE of gene expression in DG nuclei. FOS− HC and FOS− 1 h cluster into the baseline signature. FOS+ 1 h nuclei cluster into the early signature, and ARC+FOS− from both 4 and 5 h cluster into the late signature. **e** Count of all DEGs from the comparison between HC and either FOS+ 1 h, ARC+FOS− 4 h, or ARC+FOS− 5 h nuclei. Genes are grouped into categories based on temporal expression pattern. Y-axis: mean standardized expression across all genes for the given category. **f** Violin plots for genes representative of the key temporal groups of early, sustained, and late. **g** Fluorescent in situ hybridization for two genes present in the late signature, *Sorcs3* and *Blnk*, combined with immunohistochemical detection of ARC protein. Left side: 5 h post NE exposure. Right side: 1 h post NE exposure. Scale bar = 50 μm

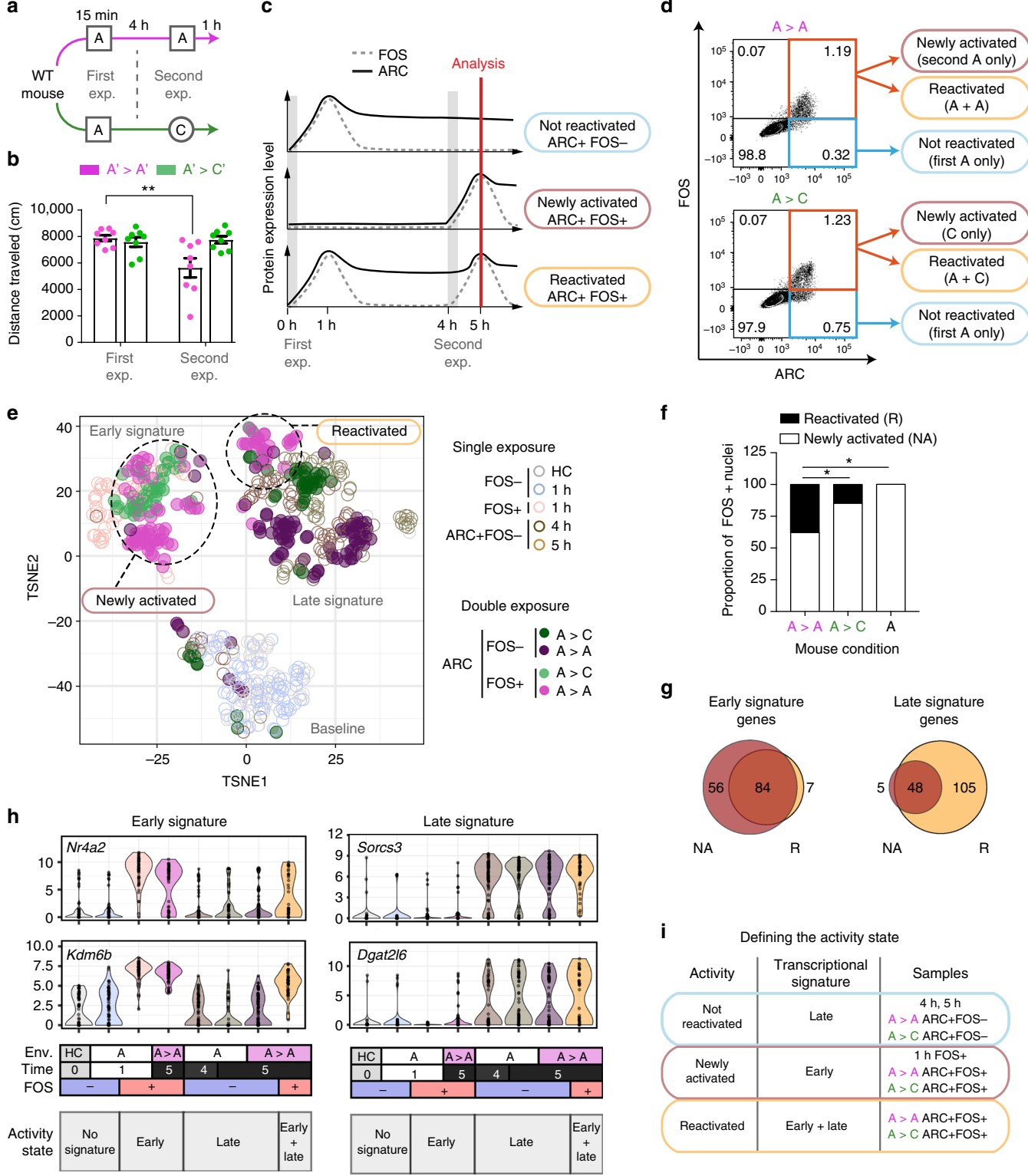

behavior, demonstrating the novelty of the second environment (Fig. 5b). These behavioral results indicated that mice maintained a memory of a first NE at the time of a second exposure 4 h later.

Based on these observations, we used the same timeline and exposed mice to A > A or A > C, two highly different environments designed to maximize IEG expression and ensure sufficient active DG neurons for subsequent analysis. The hippocampus was dissected 1 h after the second exposure, and DG neurons

were examined for activation markers using flow cytometry. Because FOS is present by 1 h and disappears by 5 h, whereas ARC has high overlap with FOS at 1 h but persists through 5 h (Fig. 5c), we classified nuclei that were ARC+FOS− as Not Reactivated (Fig. 5d, blue gate). These nuclei acquired ARC in response to the first NE but did not show the FOS signal indicative of a response to the second. The Not Reactivated population was more prominent in the A > C condition (29.2%

**Fig. 5** Early and late signatures discriminate Reactivated from Newly Activated neurons. **a** WT mice were exposed to NE A for 15 min, returned to HC for 4 h, and re-exposed to A or to a different environment C. **b** Distance traveled during a 15-min NE for mice exposed twice to the same environment or to two different environments (n = 8 per group, mean ± S.E.M., **p < 0.01, two-way ANOVA with Sidak's multiple comparisons test). **c** One hour after the second exposure, FOS and ARC protein levels alone cannot discriminate Newly Activated from Reactivated DG neurons. **d** One hour after the second exposure, FOS and ARC in DG neurons (PROX1+CTIP2+) were measured by flow cytometry. ARC+FOS+ DG neurons (red gate) = Newly Activated and Reactivated; ARC+FOS− DG neurons (blue gate) = Not Reactivated. Representative FACS plots of n = 6 mice. **e** T-SNE of all DG nuclei from the HC and single exposure (open circles) or double exposure (filled circles). Clusters that represent the baseline, early, and late signatures are noted in gray text. The group of ARC+FOS+ nuclei that clusters with the early signature is designated as putatively Newly Activated; and those that cluster with the late signature are designated as putatively Reactivated. **f** The proportion of Newly Activated and Reactivated nuclei were calculated for each exposure condition. * = Chi-square test p < 0.05. **g** Euler diagram of all early (left) and late (right) signature genes that are increased compared to HC in either the Newly Activated (NA) or Reactivated (R) nuclei. **h** Violin plots of representative genes for the early and late signatures. Env = context the mouse was exposed to, time = time in hours between exposure to the first context and sacrifice, FOS = FOS protein status by FACS, activity state = gene signature. **i** Summary of terminology. Newly Activated nuclei exhibit an early transcriptional signature, Not Reactivated nuclei exhibit the late signature, and nuclei that are activated in both contexts are expected to display both the early and late signatures (Reactivated)

(SD ± 7.4%) vs 17.1% (SD ± 5.6%) in A > A) (Supplementary Fig. 5f), which was expected based on the hypothesis that different environments should activate fewer overlapping ensembles of neurons than re-exposure to the same environment. Among the ARC+FOS+ population, we could not discriminate between Reactivated and Newly Activated nuclei based on protein signal alone (Fig. 5c). However, we hypothesized that we could distinguish between Reactivated and Newly Activated nuclei based on their transcriptomes. We collected nuclei from both ARC+FOS+ (Reactivated + Newly Activated) (Fig. 5d, red gate) and ARC+FOS− (Not Reactivated) (Fig. 5d, blue gate) populations for snRNA-seq.

All DG neurons were plotted using t-SNE (Fig. 5e). As expected, HC and 1-h FOS− nuclei clustered together (baseline signature; Fig. 5e), the 1-h FOS+ population separated from baseline (early signature; Fig. 5e), and the 4- and 5-h ARC+FOS− nuclei clustered in a separate signature (late signature; Fig. 5e). Nuclei identified as Not Reactivated by protein (ARC+FOS−) from both the A > A and A > C exposures clustered with the late signature, indicating that these neurons activated in the first environment but not the second. Conversely, the majority of ARC+FOS+ nuclei from the A > C exposure clustered with the early signature, providing further evidence that contexts A and C activated distinct sets of DG neurons. Interestingly, the ARC+FOS+ nuclei from the A > A exposure displayed two distinct signatures. One group clustered with the early signature whereas the other clustered with the late signature (Fig. 5e). This distinction potentially represented neurons that were Newly Activated (early signature) by the second experience and neurons that were Reactivated (FOS+ and late signature). There was a greater proportion of putatively Reactivated nuclei in the A > A group compared to the A > C group (Chi-square test, p-value < 3.1e−03; Fig. 5f). These Reactivated neurons from the A > A context represented possible engram cells.

We further examined the early and late gene signatures within these putative Newly Activated and Reactivated clusters from the A > A condition. IEGs, defined here as genes that were upregulated at 1 h and returned to baseline by 4 h (e.g., *Nr4a2* and *Kdm6b*; Fig. 5g–h and Supplementary Fig. 6a), were expressed at baseline levels in Not Reactivated nuclei but upregulated in Newly Activated and Reactivated nuclei. The Reactivated group expressed a smaller subset of IEGs and at a lower level of expression than the Newly Activated group (proportion $_{NA}$ = 25.0% proportion $_R$ = 10.8%; Chi-square test p-value < 3.85e−12), indicating that the second round of activity might cause weaker IEG expression in general, as previously suggested for FOS after multiple exposures[49,50]. We next examined the presence of the late signature genes in both groups. Late signature genes were primarily expressed in Reactivated

neurons (proportion $_{NA}$ = 9.46% proportion $_R$ = 18.17%; Chi-square test p-value < 9.29e−06) (e.g., *Sorcs3* and *Dgat2l6*; Fig. 5g–h and Supplementary Fig. 6a). Together these results indicated that ARC+FOS+ nuclei from the A > A condition split into two groups, one containing only a signature of recent activity (Newly Activated) and another containing signatures of recent and previous activity (Reactivated). Therefore, using FOS and ARC staining in combination with single-nucleus RNA-sequencing, we identified reactivated neurons from wild-type mice (Fig. 5i).

**Signatures established over 4 h select reactivated neurons.** We sought to determine if a component of the transcriptional signature in Reactivated nuclei was predictive of neuronal reactivity. Our strategy was to (i) identify genes that separated Reactivated from Not Reactivated neurons, (ii) select for genes that were already present 4 h after the first context, and (iii) estimate whether this signature was predictive of reactivity (Fig. 6a). A total of 884 genes were differentially expressed between Reactivated and Not Reactivated DG (FDR < 0.05; Supplementary Data 3). These reactivity DEGs were characterized by downregulation of mitochondrial genes and upregulation of kinases (Supplementary Fig. 6b). As expected, many reactivity DEGs were IEGs and were removed from further analysis. DEGs that were present at the 4- or 5-h time point and were either induced by activity (putative predictive—activity induced) or present at all time points (putative predictive—baseline) were defined as the set of putative predictive genes. Importantly, transcription of putative predictive genes exhibited notable bimodality in nuclei from mice exposed to a single context and a unimodal expression pattern in Reactivated nuclei, indicating that an underlying transcriptional signature was likely present in a subset of DG nuclei prior to entering the second context. To determine the predictive power of these genes, all A > A nuclei were separated into a training (N, $_{Newly\ Activated}$ = 15, N, $_{Not\ Reactivated}$ = 15, N, $_{Reactivated}$ = 15) and test set (N, $_{Newly\ Activated}$ = 37, N, $_{Not\ Reactivated}$ = 50, N, $_{Reactivated}$ = 21) and the ability to predict reactivity status was assessed using a random forest classifier trained on the set of putative predictive genes. A receiver-operating characteristics (ROC) curve showed that the model successfully called Reactivated neurons with an area under the curve of 0.93 and 0.96 when compared to either the Newly Activated (model i) or Not Reactivated (model ii) nuclei, respectively (Fig. 6c), and when the model was tasked with distinguishing all three groups simultaneously, classification errors of 13%, 20%, and 20% were calculated for Reactivated, Not Reactivated, and Newly Activated nuclei, respectively.

The reactivity signature was then examined in a second, independent cohort of mice exposed to A > A. A continuous value for ground truth of reactivity state was determined by identifying

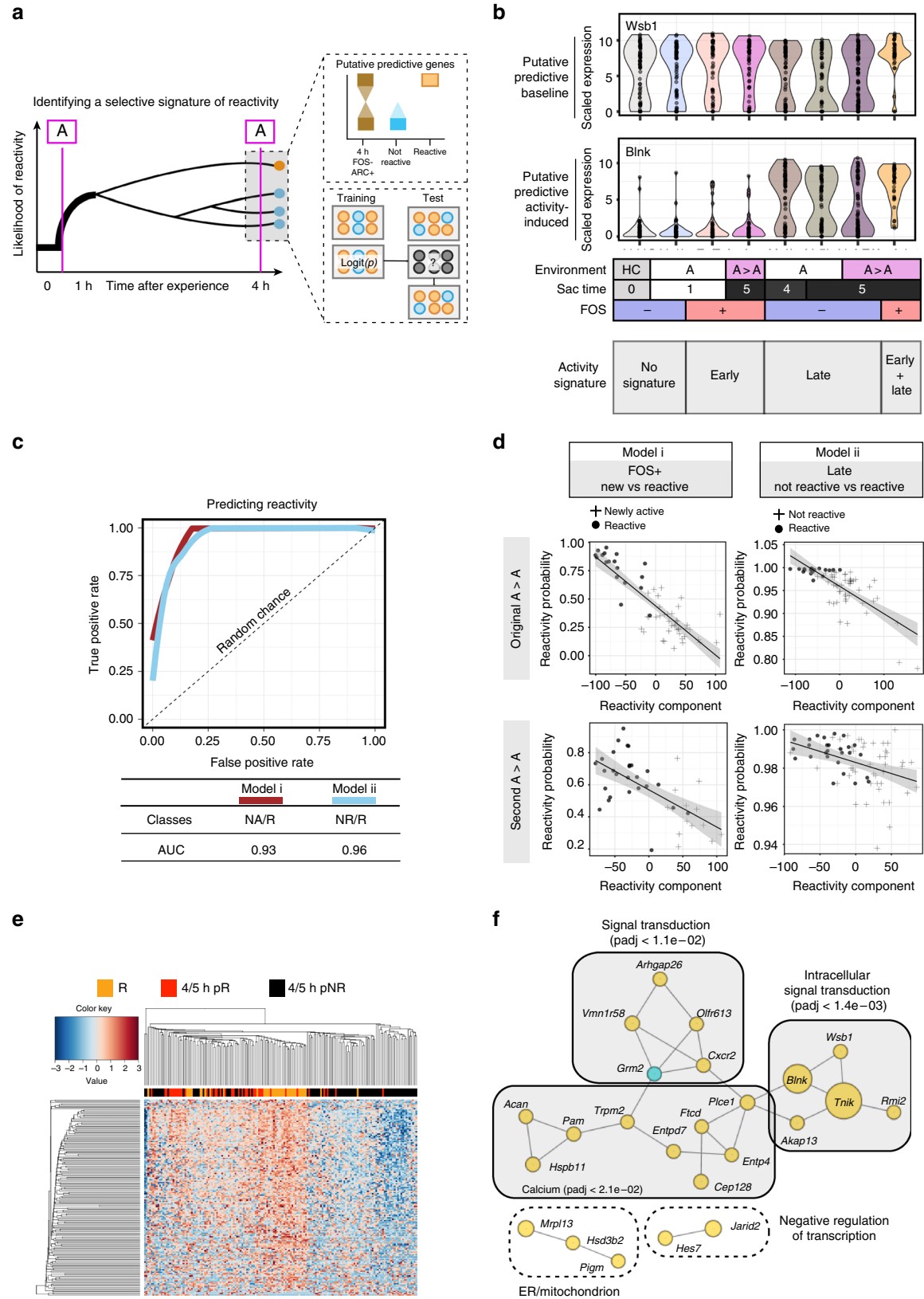

the top principal component (reactivity component) that separated either (i) FOS+ nuclei into late and early signature clusters (Supplementary Fig. 6e) (Student's $t$-test: A > A $_{original}$ $p$ < 5.20e−16; A > A $_{second}$ $p$ < 3.44e−11) or (ii) late signature nuclei into FOS+ and FOS− clusters (Student's $t$-test: A > A $_{original}$ $p$ <

1.77e−09; A > A $_{second}$ $p$ < 9.22e−09) based on the full transcriptome (Supplementary Fig. 6f). The models that were trained on the original cohort were then used to calculate the probability of reactivity in the second cohort. Using the original A > A cohort as a test case, the correlation between the reactivity component

**Fig. 6** Computational prediction of activity state. **a** Schematic of analytical procedure. (Left) Mice were exposed to two contexts (A > A). It was hypothesized that multiple transcriptional signatures developed over the intervening hours and that one of these states was associated with Reactivity. (Top right) Differential expression tests were used to identify genes that were differentially expressed between Reactivated and Not Reactivated nuclei. In order to be biologically predictive, these genes were expected to be expressed bimodally at 4 h and not in the Not Reactivated nuclei. (Bottom right) All DG nuclei from the A > A context were separated into a training and test set. The training set was used to build a classifier, which then predicted reactivity status within the test set. **b** Temporal profiles and representative violin plots of reactivity DEGs present at baseline or in late-expressing genes. **c** Model assessment using a receiver-operating characteristics curve based on predictions from the test set from the original A > A cohort. Red line = model i, blue line = model ii. The area under the curve (AUC) for each model is included in the table. **d** Model assessment for model i (first column) and model ii (second column) based on linear correlation for the original (top row) and second (bottom row) A > A cohort. Reactivity component is the principal component associated with reactivity for the given sample set. **e** Heatmap of all Reactivity DEGs showing the 4 and hour nuclei that are predicted to contain the Reactivated signature (4/5 h pR), the Not Reactivated signature (4/5 h pNR), and the true Reactivated neurons (R). **f** Predictive genes clustered by GO annotation. P-value = DAVID bioinformatics Benjamini p-value. Genes with elevated (yellow) and lower (blue) expression in Reactivated compared to Not Reactivated nuclei are shown with increasing size based on the model importance estimate. Annotations represent the top GO-term for the associated gene cluster

and the predicted reactivity was significant for both model i (Pearson's correlation test: A > A $_{original}$ $p < 3.43e-17$) and model ii (Pearson's correlation test: A > A $_{original}$ $p < 4.41e-12$), as expected. Importantly, the correlation between the reactivity component and predicted reactivity was also significant for the independent cohort of A > A nuclei (Pearson's correlation test: model i: A > A$_{second}$ $p < 2.54e-03$; model ii: A > A $_{second}$ $p < 4.57e-04$), indicating that the model was robust across multiple mice and batches (Fig. 6d). Together these findings indicated that this subset of genes had high predictive capacity for the identification of neurons with the potential to be reactivated.

This model could be applied to determine the presence of a primed state in the absence of a second exposure. We used this model to predict the reactivation potential of neurons from the 4- and 5-h time points after just a single NE exposure. Importantly, a subset of 4- and 5-h nuclei did indeed contain the predictive signature of reactivity. Clustering showed a distinct separation between the 4- and 5-h nuclei that were predicted to become reactivated and those that did not (Fig. 6e). This finding indicated that the gene expression signature associated with reactivity was present in DG neurons prior to exposure to the second context. This highly predictive set contained genes with known function in CREB and FOS pathways (e.g., *Blnk*), dentate gyrus-dependent pattern separation (*Tnik*), calcium binding (e.g., *Acan, Entpd4*), and repression of transcription (e.g., *Hes7*) (Fig. 6f). Together these results show that activity in DG neurons elicits multiple waves of transcription over time and that a key transcriptional signature is selectively enriched within potential engram cells that are reactivated upon a second exposure.

## Discussion

We identified activity-induced transcriptional changes in single DG neurons that predicted the reactivation of engram cells. Activated DG nuclei had a much stronger transcriptional change than other hippocampal cell types, possibly reflecting a major remodeling of sparsely active cells. CA1 neurons activated transcription to the same NE stimulus but to a lesser degree. Although VIP interneurons were most active based on FOS staining, they were strikingly non-responsive transcriptionally. It is known that GABAergic neurons do not express some IEGs; however, these findings were restricted to a small set of genes (e.g., *c-Fos, Arc,* and *Egr1*)[51–53]. A subsequent study examining VIP interneurons in the visual cortex showed increased transcription of experience-dependent genes when dark-housed mice were exposed to light[54]. One possible explanation for the discrepancy is that a complete lack of incoming visual sensory input may have reduced background activity levels beyond what we were able to achieve in the HC condition, pointing to a potential limitation in defining activity with FOS. Some neurons may activate a transcriptional

cascade independent of FOS. Another possible explanation is that Mardinly and colleagues[54] used a ribo-tagging method to enrich for actively translated mRNAs. VIP cells might not show widespread changes in the total number of transcripts but rather might accomplish most activity-regulated changes at the post-transcriptional level. Despite these methodological differences, our current results indicate a fundamental difference in activity-regulated processes in DG, CA1, and VIP cells. The common perception of the DG as a quiet, sparsely active region is accurate in describing a DG neuron's electrophysiological responses but not in describing its transcriptional responses.

DG neurons uniquely upregulated many genes, some of which are important for neuronal survival (e.g., *Atf3, Inhba, Bdnf, Gadd45b*)[55]. Since DG neurons are rarely active, the upregulation of survival genes may reflect enhanced protection against deleterious effects of activity such as excitotoxicity and DNA damage[55–57]. Interestingly, these neuroprotective genes also play functional roles in synaptic plasticity and adult neurogenesis (e.g., *Inhba, Bdnf, Gadd45b*), thereby providing support for additional synaptic and cellular reorganization[58–60]. It is possible that this enhanced response of synaptic plasticity genes serves to enable high fidelity encoding from sparse activity in the DG.

The early activity-induced signature in DG contained genes that facilitate nuclear changes via DNA demethylation (e.g., *Gadd45b, Gadd45g, Tet3*), histone demethylation (e.g., *Kdm6b, Kdm7a, Jmjd1c*), and transcription (e.g., *Crem, Jund, Nr4a2*). This finding is in line with previous evidence suggesting that neural activity is associated with a dramatic shift in epigenetic state[61–64] and that these epigenetic changes are important for modulating activity-dependent neuron survival and homeostatic scaling[64–66]. Importantly, a subset of these genes continued to express for up to 5 h (*Jun, Atf3, Gadd45b, Tet3*), indicating that nuclear changes continued for several hours following an initial activation event and had the potential to continue impacting memory encoding.

The late response signature in DG nuclei marked a shift from regulatory to effector genes. Proteins that influence neurite outgrowth, such as the Slitrk family (e.g., *Slitrk1, Slitrk3, Slitrk4*) and secreted extracellular matrix proteins (*Acan, Ncan, Pxdn*), were upregulated[67,68]. Some of these long-term genes hinted at a developing hypo-excitable state in DG cells. For example, late genes include Rapidly inactivated A-type potassium channels (*Kcnc4* and *Kcna4*) and *Kcnk1*, an inwardly rectifying channel that can reduce excitability[69]. *Grm7*, a metabotropic glutamate receptor that decreases evoked responses and is required for long-term depression in the dentate[70], is also upregulated late, along with the GABA$_A$ receptor α4 subunit (*Gabra4*), which mediates tonic inhibition in DG neurons[71]. This transcriptional signature of reduced excitability is intriguing in light of previous findings from Deng et al.[7] showing that, by 72 h, DG neurons were less

likely to reactivate during a second, similar exposure than at baseline. Perhaps the inhibitory transcriptional profile observed here impacts future excitability contributing to the ability of the DG to select distinct sets of neurons to encode different events.

Despite a developing gene signature that might support future inhibition, we saw robust reactivation of DG engram cells in mice re-exposed to the same NE at the 4-h time point. Importantly, the heterogeneous 4–5 h signature contained a set of genes capable of predicting the reactivity potential following just a single NE exposure. This gene set might hold valuable information as to what enables the reactivity of a DG neuron in the hours following the first activation. For example, *Tnik*, a top reactivity predictor, is a kinase that is important for synaptic and nuclear signaling pathways. Tnik is particularly important in the DG, as knockout animals display poor performance on DG-dependent spatial discrimination tasks[72]. Another top reactivity predictor was *Blnk*, whose function in neurons is largely unknown, but in B cells it is a key mediator of MAPK signaling, CREB phosphorylation, and promotion of AP1 signaling[73–75]. Although no single gene is reliably predictive of reactivation, the coordinated upregulation of multiple genes suggests that a subset of neurons is set up to overcome the overall trend toward inhibition. Together with structural changes at synapses, these transcriptional changes may be important for selecting the components of the original activated network that are ultimately involved in the encoding and retrieval of memories.

## Methods

**Contact for reagent and resource sharing**. Further information and requests for resources and reagents should be directed to the lead contact Fred H. Gage (gage@salk.edu).

**Experimental model and subject details**. Animals and treatment: All animal procedures were approved by the Institutional Animal Care and Use Committee of The Salk Institute for Biological Studies and were conducted in accordance with the National Institutes of Health's Guide for the Care and Use of Laboratory Animals.. Wild-type female C57BL/6 mice (8 weeks old) were purchased from Envigo (formerly Harlan) or Taconic and group housed (2–5 mice per cage) under standard 12-h light/dark cycles with free access to food and water. The dimensions of NE cage A were 54″ × 34″ base, 12″ height. For flow cytometry experiments, environment A included huts and tunnels that the animal was not previously exposed to. NE cage C was square rather than rectangular (42″ × 42″ base, 12″ height) and contained a different bedding material (corn cob rather than sawdust) and objects (plastic wall inserts) to distinguish it from NE cage A. NE exposures were performed between 7:00 am and 12:00 pm during light cycle. Animals did not undergo any special treatment prior to NE exposure. Mice received no exposure (HC), a single exposure to A for 15 min, or two 15-min exposures (A > A or A > C) spaced 4 h apart. Assignment of mice to groups was performed randomly. Dissections of hippocampal tissue were performed at 1 h after the final NE exposure following anesthetic overdose and cervical dislocation. Environments A and C were designed to maximize IEG expression to ensure that we could isolate sufficient active nuclei for snRNA-Seq from an extremely sparsely active population. To that end, we (1) exposed mice to the environments socially along with the their cage mates, (2) provided tunnels, huts, and running wheels that the mice could climb through and interact with and (3) used two different loose bedding types with different scents. All of these factors, while increasing the interactive nature of the environment, also create multiple moving objects that prevent us from obtaining reliable motion-tracking data; thus for behavioral characterization the environments were modified.

**Tracking of locomotion during NE exposures**. To enable tracking of locomotion in the NE, environments A and C were modified to accommodate the view of an overhead camera during the exploration period. Both environments were 54″ × 34″ base, 12″ height. The floor material was changed from bedding to a perforated metal floor (A′) or a foamboard floor (C′). Mice were singly housed and handled for 2 days prior to NE exposure. The first exposure was performed between 7:00 and 10:00 am, and the second exposure was performed between 11:00 am and 2:00 pm, both during the light cycle. Distance traveled was tracked for individual mice using the Ethovision XT software.

**Method details**. Nuclei dissociation: The nuclei dissociation protocol is similar to the previously described protocol[33], with some modifications. Hippocampus was carefully excised and immediately placed into a nuclei isolation medium (sucrose 0.25 M, KCl 25 mM, MgCl₂ 5 mM, TrisCl 10 mM, dithiothreitol 100 mM, 0.1%

Triton, protease inhibitors). Tissue was Dounce homogenized, allowing for mechanical separation of nuclei from cells. The nucleic acid stain Hoechst 33342 (5 µM, Life Technologies) was included in the media to facilitate visualization of the nuclei for quantification but excluded for sorting. Samples were washed, resuspended in nuclei storage buffer (0.167 M sucrose, MgCl₂ 5 mM, and TrisCl 10 mM, dithiothreitol 100 mM, protease inhibitors) and filtered. Solutions and samples were kept cold throughout the protocol. For RNA-seq experiments, tools and solutions were made RNAse-free and RNAse inhibitors were used (Ambion #AM2684 at 1:1000 in both isolation and storage buffers).

**Slice immunohistochemistry**. Mice were deeply anesthetized with a cocktail of ketamine/xylazine/acepromazine and transcardially perfused with 0.9% NaCl followed by 4% paraformaldehyde. Brains were removed, post-fixed overnight, and transferred to 30% sucrose for 2 days. Forty micrometer coronal sections spanning the anterior–posterior extent of the hippocampus were sectioned on a microtome and stored at −20 °C until staining. For images of VIP reactivity, immunostaining for VIP was performed with a polyclonal rabbit anti-VIP primary antibody (#20077, Immunostar, 1:1000) and donkey anti-rabbit Cy3 secondary antibody (Jackson ImmunoResearch #711-165-152, 1:250). Hippocampal cell fields were identified with antibodies against PROX1 (mouse monoclonal #MAB5654, EMD Millipore, 1:500) and CTIP2 (rat monoclonal #ab18465, Abcam, 1:200) combined with donkey anti-mouse AF488 (Jackson ImmunoResearch #715-545-151, 1:250) and donkey anti-rat AF647 (Jackson ImmunoResearch #712-605-153, 1:250). Images of IEG expression used guinea pig anti-ARC (#156005, Synaptic Systems, 1:2000) and goat anti-c-FOS (sc-52-G, Santa Cruz, 1:250) primaries and donkey anti-guinea pig Cy3 (#706-165-148, Jackson ImmunoResearch, 1:250) and donkey anti-goat AF488 secondaries (#705-545-147, Jackson ImmunoResearch, 1:250). Nuclei were visualized using DAPI (1.0 µl/ml). For images of the full hippocampus, sections were stained with antibodies against PROX1 (rabbit polyclonal #ab101851, Abcam, 1:500), CTIP2 (as above), and NEUN (mouse monoclonal conjugated to AF488 #MAB377X, EMD Millipore, 1:200). Secondary antibodies were donkey anti-rabbit Cy3 and donkey anti-rat AF647 (as above). Sections were mounted on #1.5 glass coverslips using PVA-DABCO mounting media. Confocal images were acquired on a Zeiss LSM 780 or LSM 710 laser scanning confocal microscope. VIP images were obtained using a 40× objective. ARC and FOS images were obtained using a 20× objective, and a z-stack was maximally projected for quantification and display. A blinded observer counted IEG+ cells over four maximum projections per mouse and 4–5 mice per time point. Full hippocampus images were obtained using a 20× objective, tiles were stitched using the ZEN2011 software, and a single XY plane was chosen for figure display.

**Multiplex in situ hybridization with RNAscope**. For in situ hybridization experiments, mice were perfused as above, and brains were post-fixed in 4% PFA for 24 h at 4 °C before being transferred to 30% sucrose. Coronal sections 12 µm thick were collected using a cryostat and were stored at −80 °C. In situ detection of *Sorcs3* and *Blnk* was performed using RNAscope Multiplex Fluorescent Reagent Kit v2 (Advanced Cell Diagnostics) according to the manufacturer's instructions for fixed frozen tissue. Probes used were Mm-Blnk (cat# 300031) and Mm-Sorcs3 (cat# 473421). After RNA detection, sections were co-stained for ARC protein using guinea pig anti-ARC primary (#156005, Synaptic Systems, 1:1000) and donkey anti-guinea pig horseradish peroxidase (#706-035-148, Jackson ImmunoResearch, 1:250) secondary antibodies, with TSA plus Cy5 used for fluorescent signal amplification (NEL745E001KT, PerkinElmer, 1:1000). Nuclei were visualized using DAPI (1.0 µl/ml), and sections were coverslipped in ImmuMount mounting media with #1.5 glass coverslips. Single plane images were acquired on a Zeiss Airyscan 880 microscope in confocal mode using a 20× objective.

**Flow cytometry**. To immunostain for single nuclei sorting, dissociated nuclei were incubated with mouse IgG2b anti-PROX1 (4G10, EMD Millipore, 1:300), rat IgG2a anti-CTIP2 (25B6, Abcam Ab18465, 1:300), and goat anti-c-FOS (sc-52-G, Santa Cruz, 1:500). Samples were then stained with the following secondary antibodies: donkey anti-mouse-PE (Jackson ImmunoResearch, 1:400), donkey anti-rat-Dylight 405 (Jackson ImmunoResearch, 1:400), and donkey anti-goat-AF647 (Jackson ImmunoResearch, 1:400). Centrifugation of secondary-stained samples was followed by re-suspension in mouse IgG1 anti-NEUN-AF488 (A60, EMD Millipore, 1:200). For experiments on the long-term activity signature and reactivation, no NEUN staining was performed. Instead, active DGCs were identified with PROX1, CTIP2, and c-FOS as above, and the presence of ARC was assessed with a guinea pig anti-Arc antibody (#156005, Synaptic Systems, 1:1000) combined with a donkey anti-guinea pig Cy3 secondary (Jackson ImmunoResearch, 1:400). Data from labeled samples were acquired using a BD™LSRII cytometer (BD Biosciences) followed by analysis using the FlowJo software (Tree Star). Nuclei were gated first using forward and side scatter pulse area parameters (FSC-A and SSC-A) excluding debris, followed by exclusion of aggregates using pulse width (FSC-W and SSC-W). The NEUN+ population was gated before gating populations based on PROX1, CTIP2, and FOS fluorescence (isotype control staining was used to set PROX1 and CTIP2 gates). For collection prior to amplification, a BD Influx™ sorter was used to isolate nuclei, with PBS for sheath fluid; nuclei were sorted with an 85-µM nozzle at 22.5 PSI sheath pressure. Single nuclei were directly deposited

into individual wells containing 2 μl lysis buffer in 384-well plate format. For single nuclei sorting, Single Cell (1 drop) sort mode was selected for counting accuracy. Fine mechanical alignment of the sorter's plate module was facilitated by sorting 20 10-μM fluorescent beads onto the surface of transparent plate sealer (adhered to a 384w plate), making positional adjustments as necessary; 20 beads were then sorted directly into the bottom of various well positions throughout an empty plate to provide visual confirmation of counting and targeting precision.

**Single-nuclei library preparation and sequencing**. Library preparation followed the Smart-seq2 protocol for 967 single-nuclei[32,40]. Briefly, 384w plates containing sorted single nuclei in lysis buffer were processed as follows: (1) Reverse transcription with ProtoScript II (New England Biolabs, #M0368X), (2) PCR pre-amplification with 2 × KAPA Hifi HotStart ReadyMix (Kapa, #KK2602), (3) Bead clean up with Agencourt AMPure XP beads (Beckman Coulter, #B37419AA), (4) QC with Bioanalyzer (Agilent Technologies) followed by cDNA quantification with Picogreen (LifeTech) and normalization at 0.3 ng/μl, (5) Tagmentation, indexing, and PCR amplification using Nextera XT DNA library prep kit (Illumina, #FC-121-1031), (6) Library pooling in 48-sample pools and purified with Agencourt AMPure XP beads, (7) QC of DNA libraries using D1000 Screen Tape Assay (Agilent Technologies), (8) amplifications. Sequencing was performed at the Salk Institute Next Generation Sequencing Core on an Illumina HiSeq 2500 high-throughput sequencing system with single-end 50 bp reads.

**Sequencing alignment and quality control**. Reads were trimmed using Solexa-QA++ dynamic trim[76] and were then aligned to the mm10 (GRCm38) reference genome with Ensembl gene annotation or to the ERCC reference using RSEM (bowtie)[77]. TPM values calculated by RSEM were log2+1 transformed. Raw counts, normalized TPM, and metadata information are provided in the GEO database. To determine cutoffs for outlier detection, all nuclei were analyzed by principal component analysis. The total aligned reads and total gene count were iteratively increased until the major components of variation were not associated with alignment depth or gene count. The final cutoffs were 100,000 total aligned reads and 4000 total genes detected. Nuclei below these thresholds were detected as outliers and removed from further analysis. A total of 967 nuclei were used for this study, with 868 (89.8%) surviving the filtering thresholds (Supplementary Data 5).

**Cell-type identification**. Cell type has been an increasingly difficult term to define given that clusters can almost always be split further or merged further. Therefore, to be highly confident of the final clusters for the downstream analysis of activity, it was important to perform clustering using tools that would incorporate decades of prior knowledge of neuronal cell types with unsupervised classification of transcriptional types. Since different clusters still retain varying levels of complexity, it was important to include in the workflow a calculation of within-group homogeneity, referred to below as precision. The full set of sequenced nuclei from the HC or 1-h condition were filtered for outliers as well as nuclei that were labeled as FOS- but exhibited elevated expression of *Arc*, indicating that they may have been recently active but were no longer expressing FOS protein. The workflow that we developed was as follows: (1) perform an unsupervised clustering (t-SNE, initial dimensions = 20, perplexity = 17, theta = 0, output dimensions = 2); (2) split into the minimum number of clusters (R stats hclust: hierarchical clustering, Euclidean distance, complete method); (3) determine top genes underlying those differences (edgeR, common, tagwise, and trended dispersion calculated); (4) use prior knowledge to identify known cell type; (5) create new reduced sample sets containing only one branch of the tree; (6) repeat steps 1–5 for each branch; and (7) refine clusters using randomForest[78]. For the purposes of this paper, steps 1–6 in the workflow were termed 'hierarchical iterative clustering' and step 7 was termed 'refinement.' Hierarchical iterative clustering was performed along each branch until the final two nodes were no longer divisible into groups that could support significant differential expression. These nodes were merged into single clusters. Step 2, refinement, was performed using a random forest classification method. First, a set of all genes with a minimum expression of 10 TPM summed over all samples was analyzed using random forest. A list of putatively predictive genes was identified by placing a cutoff of at least 1 with respect to the Gini coefficient. To further divide this gene list, another round of regression was performed. Fifty-one genes retained a Gini coefficient greater than 1 after this final round of regression. These 51 genes were the set of predictive genes that was used to refine the cell-type identities defined by iterative clustering by implementing the predict function in randomForest. Out-of-bag error estimates were used to determine the precision of the cluster cell type classifications. Using a random forest classifier trained on cell type-specific genes (*n*, genes = 51), we estimated the cluster precision of each group as the proportion of times that the prediction correctly identified the cell type (Supplementary Data 1). The cluster that represented DG neurons displayed the highest precision (100%), followed by VIP (99%), thalamus (98%), Pvalb (91%), CA1 (91%), subiculum (85%), Ivy (77%), and CA3 (73%). Using in situ data from the Allen Brain Atlas, the 51 genes were identified as being expressed in the predicted regions related to the cluster identity predicted by the random forest model.

**Functional enrichment**. Functional enrichment categorization was first performed for each cell-specific dataset using DAVID Bioinformatics[79], with the species *Mus musculus* used as reference. To further identify the functional categories that were present within the dataset independent of enrichment, we developed a suite of scripts that are available at https://github.com/saralinker/GONetwork. Briefly, these scripts calculated a distance measurement between each gene in a set based on similarities in GO terms. Similarity was calculated as a cosine distance with the number of parent terms associated with the GO network term included as weights into the distance calculation. Each distance matrix was calculated using the union of DEGs for each cell type. The corresponding network graph was plotted in Cytoscape[80].

**Activity state across cell types**. Genes that were detected as differentially expressed FDR < 0.05 between FOS+ and FOS− within the respective cell type of DG, CA1, or VIP were used in this analysis. The following strategy was performed used Monocle2[42]. First a sparse matrix dataset was generated using the new-CellDataSet function. Genes were detected with detectGenes and then the setOrderingFilter function was applied using only the activity-dependent gene set defined above. The expression family was set to negative binomial, and the dataset was normalized using estimateSizeFactors followed by estimateDispersions. The overall dimensionality was then reduced to two components using ICs analysis and the branching pattern was estimated using orderCells. This approach was first performed on DG, CA1, and VIP nuclei that were filtered for outliers based on gene count and alignment. The same approach was performed separately on all identified cell types that were filtered for outliers. Pseudotime was extracted and differences between groups were calculated using a Student's *t*-test.

**Motif analysis**. Homer[47] was used to perform the motif analysis. The following approach was followed for each temporally defined gene group. The script find-motifs.pl was used with the mouse reference sequence and the default settings of 400 bp upstream and 100 bp downstream of the transcriptional site and a length of approximately 8 bp. The top predicted motifs that were also defined as known and that passed multiple-testing correction were then re-analyzed using STAMP[48] to identify the closest known motif.

**Late, early, and reactive signatures**. Genes were grouped by dynamics using an intersection of results based on differential expression analysis. Four groups were used for the initial differential expression analyses: HC FOS−, 1 h FOS+, 4 h ARC+FOS−, 5 h ARC+FOS−. 1 h FOS+, 4 h ARC+FOS−, and 5 h ARC+FOS− were compared to HC FOS− nuclei and the intersection between all groups was compared by direct overlap. Group 1–7 genes were defined as those with an adjusted *p*-value < 0.05 for the comparison against 1 h FOS+; 5 h ARC+FOS−; 4 h ARC+FOS−; 4 h and 5 h ARC+FOS−; 1 h FOS+, 4 h, and 5 h ARC+FOS−; and 1 h and 5 h ARC+FOS−, respectively. The early signature was defined as the group 1 genes, the sustained signature was defined as group 5, and the late signature was defined as group 4. A continuous value for the degree to which a nucleus exhibited the late signature was calculated by performing principal component analysis using only the late signature genes. The position along PC1 was then used as a measure of late signature in each nucleus. Cells with PC1 values <10 were defined as early and values >10 as late. Double context ARC+FOS+ neurons were defined as Newly Activated or Reactivated if they were labeled as early or late, respectively, using this procedure.

**Predicting reactivity**. DEGs between Reactivated and Not Reactivated nuclei (FDR < 0.05) were filtered to maintain only those with expression in at least 80% of Reactivated nuclei, for upregulated genes, or at most 40% of Reactivated nuclei for downregulated genes where the cutoff for expression was placed at 1 log2(TPM+1). Genes were further excluded if they contained any association with batch effect or were differentially expressed in the 1-h vs HC comparison. The remaining 191 genes were passed through an initial round of feature elimination. A pairwise classification between Reactivated and either Newly Activated or Not Reactivated nuclei used a random forest classifier with a binomial family and 10,000 trees (R statistical package randomForest[78]). ROC curves were calculated by training on 15 nuclei per condition and testing on the remaining hold out samples, then calculating sensitivity and specificity using the R statistical package ROCR[81]. The genes with the top importance (Mean Decrease Gini > 0.4) in each of the pairwise comparisons were then pooled and the random forest procedure was used with a binomial family and 10,000 trees to classify all three conditions together. The classification error rate of this model was reported. This model was then used to predict the presence of a Reactivated gene signature using the 4- and 5-h time points using the base R stats predict function.

**Quantification and statistical analysis**. Differential expression: Differential expression tests were performed using the Reproducible Optimized Test Statistic (ROTS)[82] with FDR < 0.05. The bootstrap permutation parameter was set to 500, and the seed was set to 1234. Outliers as determined by gene count and alignment were excluded. FOS- cells with Arc expression > 2.5 log2(TPM+1) were excluded from analysis. To determine if differences in sample size between cell types was a driver of differences in the number of DEGs, a subsampling procedure was

implemented. Each cell-type dataset was randomly sampled without replacement to N,FOS+ = 10, N,FOS- = 10 and differential expression analysis, as described above, was performed on this reduced dataset. To minimize sampling errors, this procedure was repeated 5 times for each cell type with random starts for drawing.

**Heatmaps.** Heatmaps were plotted using heatplots from the R statistical package made4[83] with euclidean distances.

**IEG quantification.** ARC and FOS staining in fixed tissue sections were analyzed in GraphPad Prism 7 using a one-way ANOVA and compared to HC with Dunnett's multiple comparisons test.

**Gene overlap.** To calculate the number of early or late signature genes that overlapped with the Newly Activated (NA) and Reactivated (R) genes, we classified genes as being upregulated 1 or 4 h after activity (FDR < 0.01). We then calculated the overlap of these genes with the genes upregulated in Newly Activated or Reactivated compared to HC (FDR < 0.05).

**Locomotor activity during NE exposure.** Distance traveled was analyzed by two-way ANOVA in GraphPad Prism 7, and habituation between the first and second exposure was assessed with Sidak's multiple comparisons test.

**Data availability.** All data are available from the authors at request. RNA-seq data have been submitted into GEO under the accession number GSE98679.

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

## Acknowledgements

This research was supported by the NIH #MH095741 (F.H.G.) and #MH114030 (F.H.G.), a fellowship from the G. Harold & Leila Y. Mathers Charitable Foundation (F.H.G.), Annette C. Merle-Smith (F.H.G.), a fellowship from the Streims (S.B.L.), the Leona M. and Harry B. Helmsley Charitable Trust grant #2012-PG-MED00 (F.H.G.), the JPB Foundation (F.H.G.), EMBO Long-term fellowship (B.N.J.), the Bettencourt Schueller Foundation (B.N.J.) and the Philippe Foundation (B.N.J.). We thank the Salk Institute Core facilities in particular, the NGS Core Facility with funding from NIH-NCI CCSG: P30 014195, the Chapman Foundation and the Helmsley Charitable Trust; the Waitt Advanced Biophotonics Core Facility with funding from NIH-NCI CCSG: P30 014195, NINDS Neuroscience Core Grant: NS072031 and the Waitt Foundation; and the Flow Cytometry Core Facility with funding from NIH-NCI CCSG: P30 014195. We would like to thank Mary Lynn Gage for assistance in editing and Mark Novotny and Roger Lasken for technical help with the SmartSeq2 protocol.

## Author contributions

B.N.J., S.B.L., S.L.P. and F.H.G. conceptualized the study; B.N.J., J.J.B., S.L.P., I.S.G., C.D.S., C.F. and C.K.L. performed experiments and collected data; S.B.L. analyzed the single-nucleus RNA-seq datasets; B.L. contributed conceptually and provided technical expertise with nuclei isolation and RNA-seq preparation, S.T.S. performed initial microscopy imaging, S.J. provided supervision and funding, B.N.J., S.B.L. and S.L.P. wrote the manuscript. F.H.G. supervised the project.

## Additional information

**Competing interests:** B.L. is currently an employee at Fluidigm. The remaining authors declare no competing interests.

