## [Peer Review File · Nature Communications]

Reviewers' comments:

Reviewer #1 (Remarks to the Author):

In this paper, Jaegar et al. demonstrate that engram cells in the DG, CA subfield of the hippocampus, can be sorted into single cells. Using FACS, the authors have found that on exposure to a novel environment, DG cells show minimal increase in cfos whereas VIP cells show maximum increase. However, DG neurons show a large increase in the variety of expressed RNA, whereas VIP cells do not.

After subjecting the mouse through a memory re-activation protocol, the authors found that engram DG neurons expressed specific groups of RNA transcripts at specific time points, - between 1 and 5 hours after recall. The heterogeneous pattern of expression of these transcripts was used to build a predictive model that can ascertain the activity state of a given engram DG neuron, at a certain points in the memory process.

These findings represent a rather incremental step forward from the previous paper by the same group (Lacar et al., April 2016, Nature Communications), with respect to the methods used. However, the focus on DG granule cells and the re-activation experiments make the findings adequately novel that I believe they could be interesting to researchers examining a link between cellular transcriptional heterogeneity and behavioral memory states of a mouse. The authors' claims are also well supported by the project methodologies and data, and the statistical analyses are appropriately applied.

Along with its predecessor, this paper reinforces the view that transcriptional changes in a neuron, irrespective of IEG expression, can be used as a unique signature to detect their contribution to and role in the memory engram. Together, both papers provide testable hypotheses, along with adequate information and tools that can be combined with the engram technology to study phenomena beyond learning and memory. Barring some missing supplementary information and incorrect references, the submission has adequate information that will help a naive researcher reproduce the work.

Jaegar et al. present novel findings that should be of considerable interest to those in the field, and which are supported by their methodologies and data presented. I believe that with minor revisions this manuscript would be ready for publication in Nature Communications.

Minor notes:

1. Incorrect reference on page 4 line no. 2. The sentence references Park et al 201, a paper on Alzheimers. This is not related to neuronal excitability, which is the subject of the sentence.
2. Page 6 line no. 13. The reference "Williams et al., 2011" is not in the final reference (bibliography) list at the end of the document.
3. Page 6 line no. 16 "Although reliable markers are lacking for most hippocampal interneurons..." is a vague and partially un-true statement. It can be strengthened by specifying that, infact, 'transcriptional' markers are lacking, rather than protein markers that are regularly used for IHC.
4. Page 6 line no. 17 and 18. The sentence needs a reference to back up the claim about expression patterns in VIP neurons.
5. Supplementary tables 3 and 4, referenced in Page 10 line 9 and page 13 line 9 are missing.

Reviewer #2 (Remarks to the Author):

This is a very interesting and important piece of work. However, it requires substantial improvements before being published in a place such as Nature Communications.

Jaeger et al. describe in their manuscript the transcriptomic profile of hippocampal neurons that will reactivate upon mouse exposure to the same environment. The authors first characterize the activation of different hippocampal populations after exploration of a new environment. Second, they analyse the time-changes of these transcriptomic profiles in the dentate gyrus (DG). Third, the authors compare transcriptomic activation induced by exploration of two identical or different contexts. Finally, they define a transcriptional signature that predicts when a neuron will be reactivated if the animal is exposed again to the same context.

Although the claims are, in general, well demonstrated and technically well done, there are some major issues that should be addressed:

- Figure 1.

o As done with the rest of the markers used in flow cytometry, an immunohistochemistry against Fos should be shown, if possible, to validate the antibody used.

o How was the c-fos gate established in the homecage experiment? It is mentioned that isotype control staining was used to set PROX1 and CTIP2 gates, but not for Fos.

o The statistic test used in panel G should be clarified. Figure legend states that Mann-Whitney test was used, but more than two groups are compared.

- Figure 2.

o Including data from both Fos- and Fos+ homecage neurons in the t-SNE analysis would be valuable to represent baseline situation in these areas. Similarly, panel E would benefit from including analysis of expression in FOS+ neurons in homecage group.

o In order to validate the use of Fos antibody in flow cytometry, it would be necessary to include Fos expression violin plot in panel E.

- Figure 4.

o Authors should address the apparent contradiction between Fig4 panel B and Supplementary figure 5 panel A, in terms of number of c-fos positive nuclei.

- Figure 5.

o Authors fail to demonstrate that animals are able to differentiate context A and C. Panel B results do not probe that because contexts A and C used here are different to contexts used in the rest of the figure.

o In the text, authors claim that "the neurons initially active in context A did not reactivate when the mouse was instead placed in C". However, 70.8% of the neurons initially active in context A reactivate in C (Supp Fig 5). The experimental design, in general, would have benefited from designing context A and C more different – i.e. adding odorant cues.

o In panel F, the Euler diagram does not correspond with the numbers stated in the circles, and therefore the representation is misleading. Also, according to the numbers in the figure, it is not possible to conclude that the late signature genes were primarily expressed in the putatively reactivated set of neurons – as authors claim in the text.

- Figure 6.

o Predictive ability of this gene-set is not sufficiently demonstrated here. Nuclei from the same experiment (animals exposed to A-A) were divided into two groups, one used to train and one to test the predictive ability of the model. However, to test this, a different experimental cohort of mice should also be used.

o Since figure 5 compare neuronal activation upon two different contexts (A-C), it would be interesting to test the predictive power of the signature when two contexts are presented, if possible.

Finally, there are some minor issues that would help to understand the manuscript:

- Figure 2.

- o Colour blue is used to depict at the same time FOS- cells and neurons from CA1. Figure would be easier to interpret if a different colour was used for these two elements.
- o X-Axis in panel C is not clearly labelled.
- Figure 5.
- o Bigger size would help to interpret results in panel D.
- Figure 6.
- o Figure legend should be rewritten so that it better describes the figure –i.e. references to Groups 0-5 should be removed, since this information is not included in the figure.

Regarding the biological meaning of the discoveries, I find the theoretical implications of these findings are unclear. The work is quite descriptive and my concern is that it does not enhance enough the understanding of mechanisms of neuronal reactivation upon re-exposure to a context.

Reviewer #3 (Remarks to the Author):

In this work, authors use single nucleus RNA sequencing to probe activity-induced transcriptomic changes in hippocampal cells and observe gene expression differences in FOS+ dentate granule (DG) cells after novel cage exposure. Using several different types of algorithms for single cell RNAseq, authors show that DG cells display a specific activity-induced gene expression signature that lasts until 5h after novel environment exposure and can be reactivated upon a novel exposure to the same environment. Altogether, these results provide valuable new insights on the transcriptional signature that may enable DG cells to store the memory « engram ».

Overall, this paper strongly relies on FOS+ as a marker for cells undergoing increased synaptic neural activity in the context of a new experience. One major caveat in this work, is that it still remains to be demonstrated that the pattern of ensemble neural activity occurring in the hippocampus in response to a novel experience is restricted to FOS+ cells. This may only be the tip of the iceberg and this caveat should be discussed. It's also a general problem in the « engram » field.

A striking result of this work, is that the fraction of activity-induced FOS+ cells dramatically increases in the sparse VIP population. However, only a minimal number of DEG gene (n=3) were observed in this FOS+ cell population. This is surprising, given that FOS is a transcription factor regulating many genes. This result could point to an important limitation of the single nucleus RNAseq method, which does not capture activity-induced mRNAs localized in other cell compartments. Generally speaking, due to possible cell-type specific differences in mRNA trafficking and dynamics in subcellular compartments, single nucleus RNAseq method could allow a good detection of activity-induced genes in DG cells but maybe not in interneurons. This potential technical limitation could substantially change the key message of the paper. In fact, there is a now strong indication that interneurons are key players in the regulation of hippocampal ensemble activity and control the size of hippocampal memory engrams (see e.g. Stefanelli et al. neuron 2016). It is thus quite surprising that the transcriptomic changes in interneurons are so minimal, although FOS induction is the most robust in the different cell types. This discrepancy points to a technical limitation and really needs to be thoroughly discussed for this paper to be accepted.

Major points:

- Using immunohistochemistry prior to FACS, the authors were able to obtain pure cell populations for transcriptomic analysis. To obtain the final unbiased clusters, outliers and cells not fitting in

refined clusters were removed. Some key informations about this process are missing in the manuscript:

- o Which correlation method was used to determine the correlation with confounding variables? Which was the correlation score with optimal cutoff parameters for each of the experiments?

- o How many cells were obtained for each cell population prior to quality control? How many cells passed the cutoff quality control?

- o Details on sample's splitting using hierarchical iterative clustering are missing. How many cells were assigned to each pair of clusters on each iteration? Which markers were used for identifying cell types? How many cells per cluster were removed for further analysis due to random forest refinement?

- In home cage condition FOS+ neurons were isolated but no RNA analyses were presented in the paper. Are the genes differentially expressed in the HC condition similar to those observed after the novel environment exposure? The authors should provide this information and discuss this point in order to better discriminate the specific signature associated with the NE exposure.

- Information on the number of animals used for each transcriptomic experiment is missing. This information is only provided for the FACS to validate population's successful sorting (Figure 1D).

- Single molecule FISH experiments for some of the key genes will be a complementary information since this could allow to illustrate the different activity signatures identified in this study with a spatial resolution.

- Single molecule FISH experiments for some of the key genes will be illustrative of the different activity signatures identified in this study. Given that spatial information is lost in transcriptomic studies, it will be elegant to show hippocampal illustrations of the described cell population.

Minor points:

- In main text page 38 line 18, authors wrongly define t-SNE as an unbiased clustering technique even if after is clarified that the clustering algorithm used is some kind of hierarchical clustering based on Euclidean distances. Which one is? Ward? T-SNE is a dimensionality reduction algorithm used prior to distances calculation.

- There is a scale inconsistency between Figures 3B and 3C. Transcriptional activity state for each analysed cell subtype is plotted and Figure 3C gives a broader picture of subtypes for FOS+ cells. Nevertheless, the scale used is not properly described and the activity values for the same subtype differ between the graphs. Values are not even equivalent so cannot be a direct scale transformation. Information provided in those figures is unclear. Can the authors bring some light on the meaning of the scale presented here and explain more carefully how they obtained it?

- Would it be possible to compute in a more unsupervised manner the prediction of activity state presented in Figure 6? Using only differentially expressed genes for classification and prediction can bias the conclusions. One possibility will be to provide the classifier with the features used for PCA.

Reviewers' comments:

Reviewer #1 (Remarks to the Author):

In this paper, Jaegar et al. demonstrate that engram cells in the DG, CA subfield of the hippocampus, can be sorted into single cells. Using FACS, the authors have found that on exposure to a novel environment, DG cells show minimal increase in *cfos* whereas VIP cells show maximum increase. However, DG neurons show a large increase in the variety of expressed RNA, whereas VIP cells do not.

After subjecting the mouse through a memory re-activation protocol, the authors found that engram DG neurons expressed specific groups of RNA transcripts at specific time points, - between 1 and 5 hours after recall. The heterogeneous pattern of expression of these transcripts was used to build a predictive model that can ascertain the activity state of a given engram DG neuron, at a certain points in the memory process.

These findings represent a rather incremental step forward from the previous paper by the same group (Lacar et al., April 2016, Nature Communications), with respect to the methods used. However, the focus on DG granule cells and the re-activation experiments make the findings adequately novel that I believe they could be interesting to researchers examining a link between cellular transcriptional heterogeneity and behavioral memory states of a mouse. The authors' claims are also well supported by the project methodologies and data, and the statistical analyses are appropriately applied.

Along with its predecessor, this paper reinforces the view that transcriptional changes in a neuron, irrespective of IEG expression, can be used as a unique signature to detect their contribution to and role in the memory engram. Together, both papers provide testable hypotheses, along with adequate information and tools that can be combined with the engram technology to study phenomena beyond learning and memory. Barring some missing supplementary information and incorrect references, the submission has adequate information that will help a naïve researcher reproduce the work.

Jaegar et al. present novel findings that should be of considerable interest to those in the field, and which are supported by their methodologies and data presented. I believe that with minor revisions this manuscript would be ready for publication in Nature Communications.

We are pleased that the reviewer found our manuscript to be informative and scientifically sound in its original state. We hope that you will find the additional points adequately addressed below.

Minor notes:

1. Incorrect reference on page 4 line no. 2. The sentence references Park et al 201, a paper on Alzheimers. This is not related to neuronal excitability, which is the subject of the sentence.

We appreciate the reviewer's keen eye and have corrected this reference in our new manuscript.

2. Page 6 line no. 13. The reference "Williams et al., 2011" is not in the final reference (bibliography) list at the end of the document.

This reference is now in the final reference list.

3. Page 6 line no. 16 "Although reliable markers are lacking for most hippocampal interneurons..." is a vague and partially un-true statement. It can be strengthened by specifying that, infact, 'transcriptional' markers are lacking, rather than protein markers that are regularly used for IHC.

We understand that this was confusing, we have modified the text by eliminating the statement that many interneurons do not have known markers and instead pointing out only how we sorted VIP interneurons from the broader hippocampus population.

4. Page 6 line no. 17 and 18. The sentence needs a reference to back up the claim about expression patterns in VIP neurons.

We thank the reviewer for pointing this out. We have added two references to back up our claim.

- Rubin *et al.*, 2013 : PROX1: a lineage tracer for cortical interneurons originating in the lateral/caudal ganglionic eminence and preoptic area (PMID: 24155945).

- Miyoshi *et al.*, 2015 : Prox1 Regulates the Subtype-Specific Development of Caudal Ganglionic Eminence-Derived GABAergic Cortical Interneurons (PMID: 26377473).

5. Supplementary tables 3 and 4, referenced in Page 10 line 9 and page 13 line 9 are missing.

We have double-checked that all supplementary tables are included in the revised submission

Reviewer #2 (Remarks to the Author):

This is a very interesting and important piece of work. However, it requires substantial improvements before being published in a place such as Nature Communications.

Jaeger et al. describe in their manuscript the transcriptomic profile of hippocampal neurons that will reactivate upon mouse exposure to the same environment. The authors first characterize the activation of different hippocampal populations after exploration of a new environment. Second, they analyse the time-changes of these transcriptomic profiles in the dentate gyrus (DG). Third, the authors compare transcriptomic activation induced by exploration of two identical or different contexts. Finally, they define a transcriptional signature that predicts when a neuron will be reactivated if the animal is exposed again to the same context.

Although the claims are, in general, well demonstrated and technically well done, there are some major issues that should be addressed:

- Figure 1.

o As done with the rest of the markers used in flow cytometry, an immunohistochemistry against Fos should be shown, if possible, to validate the antibody used.

Immunohistochemistry stainings against FOS in homecage, 1h and 5h post novel environment exposure can be found in **Supplementary Figure 5B**.

o How was the c-fos gate established in the homecage experiment? It is mentioned that isotype control staining was used to set PROX1 and CTIP2 gates, but not for Fos.

The anti-cFOS antibody is a polyclonal antibody (Santa Cruz, sc-52-G), so isotype control staining cannot be used to set up the gate. The gating strategy can be confirmed by simultaneous staining of FOS and another immediate early gene in homecage. Co-staining with FOS and ARC shows that 72.4% (SD+/- 5.2, n=3) of the DG neurons present in the FOS+ gate in homecage co-express ARC (Figure 1). In addition, the number of FOS+ neurons obtained by FACS gating in homecage is comparable to the number of FOS+ neurons found by immunohistochemistry staining in the same conditions (Figure 2). Together, these results indicate that the FOS+ FACS gate accurately identifies active cells in homecage. This supporting figure has now been added as **Supplementary Figure 2C**.

Figure 1: Representative FACS plots showing the percentage of ARC+ neurons among FOS+ DG neurons isolated from homecage mice (n=3).

o The statistic test used in panel G should be clarified. Figure legend states that Mann-Whitney test was used, but more than two groups are compared.

We agree with the reviewer, we have now performed a one-way ANOVA test and used Tukey's multiple comparison test to compare the four groups of neurons. This has been clarified in the legend.

- Figure 2.

o Including data from both Fos- and Fos+ homecage neurons in the t-SNE analysis would be valuable to represent baseline situation in these areas. Similarly, panel E would benefit from including analysis of expression in FOS+ neurons in homecage group.

To answer the reviewer's comment, we have additionally sorted, sequenced, and analyzed DG nuclei that exhibit elevated levels of FOS (FOS low) from homecage and animals 1hr after exposure and have added the following paragraph to the results section:

"In the homecage a subset of DG neurons displayed FOS levels that were higher than the FOS- gate, but lower than the level of FOS in DG neurons following activity. We termed these neurons "FOS low" and examined whether activity-induced expression in FOS+ neurons was stronger than the expression in FOS low. Monocle was used to calculate the progression of activity-induced gene expression between all FOS-, FOS low, and FOS+ DG neurons (Supplementary Figure 3B). Similar to the FOS protein stain, the FOS low DG neurons from both the homecage and 1hr animals were present

in an intermediate position between FOS- and FOS+. Furthermore while top differentially expressed genes such as *Arc* and *Inhba* were increased in the FOS low cells, this expression was lower than that of FOS+ and displayed higher variability (Supplementary Figure 3C).”

Figure 2: DG nuclei expressing low levels of FOS have an intermediate level of activity-induced gene expression

Top left: FACS plot of FOS expression within DG nuclei with FOS-, FOS low, and FOS+ nuclei denoted. Bottom left: Monocle plot of FOS-, FOS low, and FOS+ DG nuclei from the homecage (HC) or 1hr context using only activity-induced genes as input. Note the linear distribution of nuclei along IC2 from FOS- to FOS low to FOS+. This indicates that FOS low nuclei have an elevated level of activity-induced genes, but to a lower threshold than FOS+ nuclei. Bottom right: Example expression of two activity-induced genes *Arc* and *Inhba* which are highest expressed in FOS+ nuclei followed by FOS low and then FOS-.

o In order to validate the use of Fos antibody in flow cytometry, it would be necessary to include Fos expression violin plot in panel D.

We have now included the violin plot of Fos expression in **Figure 2 panel E** as well as this additional statement in the results section.

“As expected all cell types exhibited higher levels of Fos RNA in the nuclei stained for FOS protein (DG: $p_{raw} < 3.45e-05$; CA1 $p_{raw} < 1.8e-04$; VIP $p_{raw} < 6.5e-03$. Figure 2E).”

Figure 3: c-Fos expression in hippocampal nuclei following exposure to a novel environment. c-Fos expression is denoted as \log_2 transcripts per million (TPM) for DG, CA1, and VIP from FOS- (blue violins) and FOS+ (red violins) neurons from the homecage and novel environment.

- Figure 4.

o Authors should address the apparent contradiction between Fig4 panel B and Supplementary figure 5 panel A, in terms of number of c-fos positive nuclei.

We understand that this was confusing. To address this point, we performed a direct comparison between cell counts obtained by FACS and immunohistochemistry stainings. We calculated cell counts obtained by FACS based on a total number of 300,000 DG neurons (Pilz et al., Science, 2018, PMID:29439238). Comparison between the number of FOS+ DG cells counted by FACS and by immunohistochemistry shows similar results both in homecage condition and after novel environment exposure (Figure 2). This figure has now been added to the manuscript as **Supplementary Figure 5C**.

Figure 4: Comparison of the total number of FOS+ DG neurons obtained by FACS and by immunohistochemistry stainings in homecage (HC) or 1h after novel environment exposure (NE 1hr). (t-test, n.s $p > 0.05$, $n=5$).

- Figure 5.

o Authors fail to demonstrate that animals are able to differentiate context A and C. Panel B results do not probe that because contexts A and C used here are different to contexts used in the rest of the figure.

The reviewer brings up an important point. It is true that we were unable to track the behavior of mice in the exact same environments used for the sorting experiments. We provided the data in panel B as evidence that when exposed to similar environments to those used for nuclei sorting, mice maintain a memory of those environments during the 4hr gap between exposures.

Environments A and C were originally designed to maximize IEG expression to ensure that we could isolate sufficient active nuclei for snRNA-Seq from an extremely sparsely-active population. To that end, we designed a highly interactive environment. We 1) exposed mice to the environments socially along with their cage mates, 2) provided tunnels, huts, and running wheels that the mice could climb through and interact with and 3) used two different loose bedding types with different scents. All of these factors, while increasing the interactive nature of the environment, also create multiple moving objects that prevent us from obtaining reliable motion-tracking data. We explored the possibility of using a contextual fear conditioning paradigm for these studies, but our pilot data revealed substantially fewer Arc+ cells after fear conditioning (FC) compared to large novel environments (NE), and we were not confident we would be able to sort and sequence sufficient nuclei without pooling tissue from multiple mice, particularly at later time points.

To clarify this point, we have added the following sentence in our revised manuscript: *“These behavioral results indicate that mice maintain a memory of the first NE at the time of the second exposure 4 hr later. Based on these observations, we used the same*

timeline and exposed mice to highly different environments (A and C) designed to maximize IEG expression to ensure that we could isolate enough active DG neurons for subsequent analysis. “

Figure 5: Arc staining by histology after exposure to homecage (HC), fear-conditioning (FC), or a novel environment (NE)

o In the text, authors claim that “the neurons initially active in context A did not reactivate when the mouse was instead placed in C”. However, 70.8% of the neurons initially active in context A reactivate in C (Supp Fig 5). The experimental design, in general, would have benefited from designing context A and C more different – i.e. adding odorant cues.

We apologize for this confusion. Supplementary Figure 5 indicates the percentage of neurons that did not reactivate among all ARC+ nuclei. This implies that 70.8% of neurons are activated (FOS+) in the second context but not necessarily Reactivated. Indeed, ARC+FOS+ neurons encompass both Newly activated (second context only) and Reactivated neurons (first and second context). To clarify this point, we have added the following figure (figure 6) to **Figure 5, panel C** in our revised manuscript.

Figure 6: Schematic representation of ARC and FOS expression in the different DG neuron populations following second environment exposure.

o In panel F, the Euler diagram does not correspond with the numbers stated in the circles, and therefore the representation is misleading. Also, according to the numbers in the figure, it is not possible to conclude that the late signature genes were primarily expressed in the putatively reactivated set of neurons – as authors claim in the text.

We thank the reviewers for pointing out this error and have correct the representation of the euler diagrams. With respect to the conclusion we have also added a test of the significance of the difference in proportion of the overlap of newly activated and reactivated genes expressing either the early or late signature genes using a 2-sample test for equality of proportions. The genes upregulated in the newly active nuclei overlap with a larger proportion of early signature genes (25.0%) than the genes upregulated in the reactivated nuclei (10.8%) (p -value < $3.85e-12$). Conversely the newly activated nuclei express fewer of the late signature genes (9.5%) when compared to the reactivated nuclei (18.2%) (p -value < $9.29e-06$).

These tests have been added to the manuscript.

- Figure 6.

o Predictive ability of this gene-set is not sufficiently demonstrated here. Nuclei from the same experiment (animals exposed to A-A) were divided into two groups, one used to train and one to test the predictive ability of the model. However, to test this, a different experimental cohort of mice should also be used.

We understand the reviewer's concern that an additional independent cohort should be used to determine the reproducibility of the model. To address this concern we have generated an additional independent cohort of mice exposed to A>A that the reviewers have asked for (second cohort; A>A). The analysis is explained in detail in the following response which refers to assessing the reactivity in the A>C cohort.

o Since figure 5 compare neuronal activation upon two different contexts (A-C), it would be interesting to test the predictive power of the signature when two contexts are presented, if possible.

We agree that assessing the reactivity signature within the A>C cohort is interesting and will answer this question in combination with the prior comment from this reviewer where they have asked to assess the model on an independent A>A cohort. However, it should be noted that given that only 4 neurons were identified to be putatively reactivated in the A>C context, and given the expectation of different networks of neurons being activated in C compared to A, we feel that the results from the A>C comparison should be taken lightly.

Briefly, a second set of two mice were exposed to the A>A condition. As with the first cohort, DG nuclei were sorted on ARC(+) and FOS(-/+). The two models being used were: model i) Discriminated newly active nuclei (FOS+, early signature) from reactivated nuclei (FOS+, late signature) and model ii) Discriminated not reactivated nuclei (FOS-, late signature) from reactivated nuclei. In all cases the FOS status was identified by protein staining and the early / late signatures were identified by clustering on the whole transcriptome.

We assessed the probability of reactivity provided by the model as a function of a continuous variable of "ground truth". To obtain a continuous variable associated with ground truth, Principal Component Analysis was performed on nuclei from the original A>A, second A>A, or A>C cohort. We then assessed which component was associated with putative reactivity as denoted by clustering and this component was used as a continuous variable for ground truth. The continuous ground truth variable was then correlated with the probability of reactivity to assess the validity of the model.

Model i was significantly associated with the linear ground truth for both the original A>A, second A>A cohorts, but not in A>C. The probability of reactivity using model ii was also significantly correlated with linear ground truth for both the original and second A>A cohorts. However, reactivity probability was not able to be determined in the A>C group. While it is tempting to speculate that the inability to discriminate reactivity in the

A>C group indicates that the transcriptional signature is unique to the 'A' context, the sample size of reactivated nuclei in A>C is too small to make firm conclusions. For this reason we have report the results for the second A>A cohort only in **Figure 6D and Supplementary Figures 6E and F** and have included the following information to the results section.

“The presence of the reactivity signature was then examined in a second, independent, cohort of mice exposed to A>A. A continuous value for ground truth of the reactivity state was determined by identifying the top principal component (reactivity component) that separated either i) FOS+ nuclei into late and early signature clusters (Supplementary Figure 6E) ($A>A_{original} p < 5.20e-16$; $A>A_{second} p < 3.44e-11$) or ii) late signature nuclei into FOS+ and FOS- clusters ($A>A_{original} p < 1.77e-09$; $A>A_{second} p < 9.22e-09$) based on the full transcriptome (Supplementary Figure 6F). The models which were trained on the original cohort were then used to calculate the probability of reactivity in the second cohort. Using the original A>A cohort as a test case, as expected, the correlation between the reactivity component and the predicted reactivity was significant for both model i ($A>A_{original} p < 3.43e-17$) and model ii ($A>A_{original} p < 4.41e-12$). Importantly, the correlation between the reactivity component and predicted reactivity was also significant for the independent cohort of A>A nuclei (model i: $A>A_{second} p < 2.54e-03$; model ii: $A>A_{second} p < 4.57e-04$) indicating that the model was robust across multiple mice and batches (Figure 6D). ”

Figure 7: Model assessment using the raw prediction probabilities
 For each cohort (First A>A, A>C, or Second A>A) ground truth (top row) and model assessment (bottom row) are plotted for models *i* (left column) and *ii* (right column).

Finally, there are some minor issues that would help to understand the manuscript:

- Figure 2.

o Colour blue is used to depict at the same time FOS- cells and neurons from CA1. Figure would be easier to interpret if a different colour was used for these two elements.

We apologize if the FOS- and CA1 populations are very similar in color. To avoid adding another color to the particularly large panel of colors used across our manuscript, we

have increased the color contrast between the two populations in the new version of Figure 2 and hope that this facilitates the interpretation of the figure.

o X-Axis in panel C is not clearly labelled.

We have changed the x-axis so it more clearly states that the values of the bars are equivalent to the number of differentially expressed genes (DEGs). Counts in the positive direction are those that are higher in the cell group (FOS+ or FOS-) when compared to homecage. Counts in the negative direction are those that are lower in the cell group compared to homecage.

- Figure 5.

o Bigger size would help to interpret results in panel D.

This size of this panel has been increased.

- Figure 6.

o Figure legend should be rewritten so that it better describes the figure –i.e. references to Groups 0-5 should be removed, since this information is not included in the figure.

We have substantially restructured figure 6 and have included here the results to the additional comparisons requested by the reviewer. In the process we have rewritten the figure legend and have removed references to groups 0-5.

Regarding the biological meaning of the discoveries, I find the theoretical implications of these findings are unclear. The work is quite descriptive and my concern is that it does not enhance enough the understanding of mechanisms of neuronal reactivation upon re-exposure to a context.

We don't dispute that many of our results are descriptive, but we do believe our work helps to bridge a knowledge gap that still plagues studies of learning and memory. Prior studies have found that tagging active cells, then later reactivating or silencing those cells, appears to trigger or repress memory recall, respectively. These studies are undoubtedly exciting, but they also require an artificial perturbation of the neuronal network involved.

We wanted to understand the endogenous mechanisms more thoroughly. Does the presence of Fos, which has been near-universally applied throughout the brain as an activity tag, imply the same downstream consequences in different cell types? Even within the hippocampus, the stark differences between DG, CA1, and VIP cells reveal that it does not. Previous studies have suggested memory consolidation is sensitive to protein synthesis inhibitors up to 2 days after the encoding event, so what occurs in

individual active cells once IEGs such as Fos disappear? Our results identified a heterogeneous late wave of transcription 4-5hr after activity, and this heterogeneity appears to have functional consequences. We agree that much remains to be done in understanding the roles of specific genes and pathways, and we hope this work provides fertile ground for future hypothesis-driven research.

Reviewer #3 (Remarks to the Author):

In this work, authors use single nucleus RNA sequencing to probe activity-induced transcriptomic changes in hippocampal cells and observe gene expression differences in FOS+ dentate granule (DG) cells after novel cage exposure. Using several different types of algorithms for single cell RNAseq, authors show that DG cells display a specific activity-induced gene expression signature that lasts until 5h after novel environment exposure and can be reactivated upon a novel exposure to the same environment. Altogether, these results provide valuable new insights on the transcriptional signature that may enable DG cells to store the memory « engram ».

Overall, this paper strongly relies on FOS+ as a marker for cells undergoing increased synaptic neural activity in the context of a new experience. One major caveat in this work, is that it still remains to be demonstrated that the pattern of ensemble neural activity occurring in the hippocampus in response to a novel experience is restricted to FOS+ cells. This may only be the tip of the iceberg and this caveat should be discussed. It's also a general problem in the « engram » field.

A striking result of this work, is that the fraction of activity-induced FOS+ cells dramatically increases in the sparse VIP population. However, only a minimal number of DEG gene (n=3) were observed in this FOS+ cell population. This is surprising, given that FOS is a transcription factor regulating many genes. This result could point to an important limitation of the single nucleus RNAseq method, which does not capture activity-induced mRNAs localized in other cell compartments. Generally speaking, due to possible cell-type specific differences in mRNA trafficking and dynamics in subcellular compartments, single nucleus RNAseq method could allow a good detection of activity-induced genes in DG cells but maybe not in interneurons. This potential technical limitation could substantially change the key message of the paper. In fact, there is a now strong indication that interneurons are key players in the regulation of hippocampal ensemble activity and control the size of hippocampal memory engrams (see e.g. Stefanelli et al. neuron 2016). It is thus quite surprising that the transcriptomic changes in interneurons are so minimal, although FOS induction is the most robust in the

different cell types. This discrepancy points to a technical limitation and really needs to be thoroughly discussed for this paper to be accepted.

We appreciate the thoughtful insight from this reviewer. We have now added the following statement to our discussion.

“A subsequent study examining VIP interneurons in the visual cortex showed increased transcription of experience-dependent genes when mice were first dark-housed and then exposed to light {Mardinly, 2016 #172}. One possible explanation for the discrepancy is that a complete lack of incoming visual sensory input may reduce background activity levels in visual cortex beyond what we are able to achieve in the hippocampus in the homecage condition. This points to a potential limitation in defining active and inactive neurons as those which do and do not express FOS protein 1 hr following exposure to a novel environment. FOS is not the only transcription factor induced by activity, and therefore there may be neurons that have activated a transcriptional cascade independent of FOS expression. Another possible explanation is that Mardinly and colleagues used a ribo-tagging method to explicitly enrich for actively translated mRNAs. VIP cells may not show widespread changes in the total number of transcripts due to activity, but rather accomplish most activity-regulated changes at the post-transcriptional level.”

Major points:

- Using immunohistochemistry prior to FACS, the authors were able to obtain pure cell populations for transcriptomic analysis. To obtain the final unbiased clusters, outliers and cells not fitting in refined clusters were removed. Some key informations about this process are missing in the manuscript:

o Which correlation method was used to determine the correlation with confounding variables? Which was the correlation score with optimal cutoff parameters for each of the experiments?

A correlation score was not used for outlier exclusion. Outliers were determined by total aligned reads < 100,000 and total gene count < 4,000. There were additionally 6 nuclei (3 FOS- 1hr NEUN+PROX1-CTIP2+ , 1 FOS- HC NEUN+PROX1-CTIP2+, 2 FOS- HC NEUN+PROX1+CTIP2-) that were outliers when examining the first and second components in PCA space. These were excluded to increase sensitivity in clustering.

o How many cells were obtained for each cell population prior to quality control? How many cells passed the cutoff quality control?

The number of nuclei obtained for each population and the percent of those passing quality control are noted in the following table, which has now been added as Supplementary Table 5.

Population Sorted	All Nuclei											
	HC FOS-	HC FOS low	1hr FOS-	1hr FOS low	1hr FOS high	4hr	5hr	A>A ARC+FOS-	A>A ARC+FOS+	A>C ARC+FOS-	A>C ARC+FOS+	
NEUN+PROX1+CTIP2+	24 (100%)	25 (88%)	29 (93%)	30 (93%)	27 (96%)	74 (92%)	96 (61%)	142 (91%)	144 (90%)	48 (96%)	48 (100%)	
NEUN+PROX1-CTIP2+	29 (100%)	NA	28 (100%)	NA	31 (93%)	NA	NA	NA	NA	NA	NA	
NEUN+PROX1-CTIP2-	32 (100%)	NA	32 (100%)	NA	32 (100%)	NA	NA	NA	NA	NA	NA	
NEUN+PROX1+CTIP2-	33 (90%)	NA	33 (66%)	NA	30 (100%)	NA	NA	NA	NA	NA	NA	

nuclei sequenced (% nuclei analyzed)

o Details on sample's splitting using hierarchical iterative clustering are missing. How many cells were assigned to each pair of clusters on each iteration? Which markers were used for identifying cell types? How many cells per cluster were removed for further analysis due to random forest refinement?

The full set of sequenced nuclei from the home cage or 1hr condition were filtered for outliers as well as nuclei that were labelled as FOS- but exhibited elevated expression of *Arc* that indicated that they may have been recently active but no longer expressing FOS protein. The clustering procedure began with 298 nuclei which were then iteratively clustered. We did not have access to the original cell counts for each level of iteration and therefore have reproduced this analysis in the table below with no major differences from the original analysis. The predicted cell type and relevant marker gene are included for each cluster in each iteration. The cell type label assigned from hierarchical clustering and the associated count of nuclei, is marked in blue in the table below. At the end of the clustering procedure we then passed the dataset through a random forest classifier to refine the identity of each nucleus and to determine cluster precision. No nuclei were removed during random forest classification. Given the high predictive nature of the random forest classifier and the propensity for random noise in t-SNE, nuclei were reassigned to a different identify if they exhibited a strong signature that placed them in a different cell type when analyzed by random forest than the cell type identified by clustering.

Iterative Hierarchical Clustering							
Iteration	input	cluster 1			cluster 2		
		nuclei count	marker	cell type	nuclei count	marker	cell type
1	all nuclei	205	Slc17a7	Glutamatergic	93	Gad2	GABAergic
2	iteration 1 - cluster 2	36	Nxph1	GABAergic	57	Vip	VIP
3	iteration 2 - cluster 2	46	Sema5a	VIP-Sema5a	11	Sncg	VIP-Sncg
4	iteration 2 - cluster 1	28	Gad2	GABAergic	8	Tcf7l2	Thalamus
5	iteration 4 - cluster 1	10	Pvalb	Pvalb	18	Sez6	GABAergic
6	iteration 5 - cluster 2	12	Lamp5	Ivy	6	Vip	VIP
7	iteration 1 - cluster 1	129	Sv2b	Glutamatergic	76	Prox1	DG
8	iteration 7 - cluster 1	100	Kcnh7	Glutamatergic	29	Grik4	CA3
9	iteration 8 - cluster 1	73	Iqgap2	CA1	27	Tshz2	Sub

Random Forest							
CA1	CA3	DG	Thalamus	Ivy	Pvalb	Subiculum	VIP
79	28	76	8	10	9	22	66

- In home cage condition FOS+ neurons were isolated but no RNA analyses were presented in the paper. Are the genes differentially expressed in the HC condition similar to those observed after the novel environment exposure? The authors should provide this information and discuss this point in order to better discriminate the specific signature associated with the NE exposure.

While there are cells expressing FOS in the homecage condition, they express FOS at a lower level than 1hr after exposure to a novel environment. To respond to the reviewer's request we have additionally sorted, sequenced, and analyzed DG nuclei that exhibit elevated levels of FOS (FOS low) from homecage animals. FOS low neurons also exist 1hr after exposure, therefore for a clear comparison we have sorted and sequenced the FOS low state from the 1hr condition as well and have added the following paragraph to the results section:

"In the homecage, a subset of DG neurons displayed FOS at a detectable but lower level compared to that induced in DG neurons following NE exposure (Supplementary Figure 3A). We termed these neurons "FOS low" and examined whether activity-induced expression in FOS+ neurons was stronger than the expression in FOS low. Monocle was used to calculate the progression of activity-induced gene expression between all FOS-, FOS low, and FOS+ DG neurons (Supplementary Figure 3B). Similar to the FOS protein stain, the FOS low DG neurons from both the homecage and 1hr animals were present in an intermediate position between FOS- and FOS+. Furthermore while top differentially expressed genes such as Arc and Inhba were increased in the FOS low cells, this expression was lower than that of FOS+ and displayed higher variability (Supplementary Figure 3C)."

Figure 8. DG nuclei expressing low levels of FOS have an intermediate level of activity-induced gene expression

(Left) Monocle plot of FOS-, FOS low, and FOS+ DG nuclei from the homecage (HC) or 1hr context using only activity-induced genes as input. Note the linear distribution of nuclei along IC2 from FOS- to FOS low to FOS+. This indicates that FOS low nuclei have an elevated level of activity-induced genes, but to a lower level than FOS+ nuclei. (Right) Example expression of two activity-induced genes Arc and Inhba which are highest expressed in FOS+ nuclei followed by FOS low and then FOS-.

- Information on the number of animals used for each transcriptomic experiment is missing. This information is only provided for the FACS to validate population's successful sorting (Figure 1D).

We have provided this information in the table below.

Population Sorted	Mice										
	HC FOS-	HC FOS low	1hr FOS-	1hr FOS low	1hr FOS high	4hr	5hr	A>A ARC+FOS-	A>A ARC+FOS+	A>C ARC+FOS-	A>C ARC+FOS+
NEUN+PROX1+CTIP2+	2	2	1	1	1	2	4	4	4	2	2
NEUN+PROX1-CTIP2+	1	NA	1	NA	1	NA	NA	NA	NA	NA	1
NEUN+PROX1-CTIP2-	2	NA	2	NA	2	NA	NA	NA	NA	NA	NA
NEUN+PROX1+CTIP2-	2	NA	2	NA	2	NA	NA	NA	NA	NA	NA

nuclei sequenced (% nuclei analyzed)

- Single molecule FISH experiments for some of the key genes will be a complementary information since this could allow to illustrate the different activity signatures identified in this study with a spatial resolution.

We agree with this reviewer that FISH experiments are helpful to understand the validity of our analysis. We have performed FISH for two of the genes that are expressed late after exposure to an enriched environment: *Sorcs3* and *Blnk*. We observed substantial colocalization of ARC protein, *Sorcs3*, and *Blnk* 5hr after NE exposure. These results now appear in Figure 4G.

- Single molecule FISH experiments for some of the key genes will be illustrative of the different activity signatures identified in this study. Given that spatial information is lost in transcriptomic studies, it will be elegant to show hippocampal illustrations of the described cell population.

As discussed above, we chose to validate our snRNA-Seq results using two genes induced by activity in DG, *Sorcs3* and *Blnk*. Although we did not pursue CA1 or VIP-specific genes further, we did observe that *Blnk* was noticeably absent in CA1 after NE exposure despite widespread expression of *Sorcs3*, confirming that activity-induced gene expression differs across these two cell populations.

Minor points:

- In main text page 38 line 18, authors wrongly define t-SNE as an unbiased clustering technique even if after is clarified that the clustering algorithm used is some kind of hierarchical clustering based on Euclidean distances. Which one is? Ward? T-SNE is a dimensionality reduction algorithm used prior to distances calculation.

We apologize for this misnomer as there are certainly biases when using any dimensionality reduction method. The intention was to state that it was unsupervised and this has been changed accordingly in the text. The analysis was performed in R using the base R function `hclust` with the complete agglomeration method.

- There is a scale inconsistency between Figures 3B and 3C. Transcriptional activity state for each analysed cell subtype is plotted and Figure 3C gives a broader picture of subtypes for FOS+ cells. Nevertheless, the scale used is not properly described and the activity values for the same subtype differ between the graphs. Values are not even equivalent so cannot be a direct scale transformation. Information provided in those figures is unclear. Can the authors bring some light on the meaning of the scale presented here and explain more carefully how they obtained it?

With respect to the scale inconsistency between figures 3b and 3c, these figures were derived from independent calculations and therefore are not identical. In figures 3a and

b ICA was performed only on the FOS+ and FOS- cells from DG, CA1, and VIP. Conversely, in figures 3c and d ICA was performed on all FOS+ cells from all cell populations. The figure legend has now been edited to clarify this distinction.

The scale on the y-axis is pseudotime as calculated by the monocle algorithm. In brief, the activity-dependent genes were used as input into the dimensionality reduction portion of the algorithm (independent component analysis). Monocle then identifies the minimum distance between points in two-dimensions while taking into account branch structure (Figs 3a and 3c). It uses this path information to calculate a continuous variable through all of the points denoted as “pseudotime”. This is the value of the y-axis in Figures 3b and 3d. This information has now been added to the figure legend.

- Would it be possible to compute in a more unsupervised manner the prediction of activity state presented in Figure 6? Using only differentially expressed genes for classification and prediction can bias the conclusions. One possibility will be to provide the classifier with the features used for PCA.

We agree that the original manuscript would be strengthened by reducing the dependence between classification and prediction. To address this concern we have generated an independent cohort of nuclei from two additional mice exposed to A>A. We have now assessed the model using both the original A>A cohort as well as the second A>A cohort. Briefly, a second set of two mice were exposed to the A>A condition. As with the first cohort, DG nuclei were sorted on ARC(+) and FOS(-/+).

We have provided below a table summarizing the predictions of the reactivity models for the original and second A>A cohorts. The two models being used were: model i) Discriminated newly active nuclei (FOS+, early signature) from reactivated nuclei (FOS+, late signature) and model ii) Discriminated not reactivated nuclei (FOS-, late signature) from reactivated nuclei. In all cases the FOS status was identified by protein staining and the early / late signatures were identified by clustering on the whole transcriptome.

We assessed the probability of reactivity provided by the model as a function of a continuous variable of “ground truth”. To obtain a continuous variable associated with ground truth, Principal Component Analysis was performed on nuclei from the original A>A, second A>A, or A>C cohort. We then assessed which component was associated with putative reactivity as denoted by clustering and this component was used as a continuous variable for ground truth. The continuous ground truth variable was then correlated with the probability of reactivity to assess the validity of the model.

Probability assessment: Model i (newly active vs. reactive)

Model i was significantly associated with the linear ground truth for both the original and second A>A cohorts. This indicated that there was a clear distinction between newly active and reactivated nuclei using model i.

Probability assessment: Model ii (not reactivated vs. reactive)

The probability of reactivity using model ii was significantly correlated with linear ground truth for both the original and second A>A cohorts.

We have provided the results for the second A>A cohort in the manuscript as Figure 6 D and Supplementary Figures 6E and F and have included the following information to the results section.

“The presence of the reactivity signature was then examined in a second, independent, cohort of mice exposed to A>A. A continuous value for ground truth of the reactivity state was determined by identifying the top principal component (reactivity component) that separated either i) FOS+ nuclei into late and early signature clusters (Supplementary Figure 6E) ($A>A_{original} p < 5.20e-16$; $A>A_{second} p < 3.44e-11$) or ii) late signature nuclei into FOS+ and FOS- clusters ($A>A_{original} p < 1.77e-09$; $A>A_{second} p < 9.22e-09$) based on the full transcriptome (Supplementary Figure 6F). The models which were trained on the original cohort were then used to calculate the probability of reactivity in the second cohort. Using the original A>A cohort as a test case, as expected, the correlation between the reactivity component and the predicted reactivity was significant for both model i ($A>A_{original} p < 3.43e-17$) and model ii ($A>A_{original} p < 4.41e-12$). Importantly, the correlation between the reactivity component and predicted reactivity was also significant for the independent cohort of A>A nuclei (model i: $A>A_{second} p < 2.54e-03$; model ii: $A>A_{second} p < 4.57e-04$) indicating that the model was robust across multiple mice and batches (Figure 6D). ”

Figure 10: Independent assessment of reactivity predictions

A) (Top) T-SNE of second cohort of A>A mice colored by FOS stain (F = FOS+, N = FOS-). (Bottom) Expression of the late-expressed gene *Sorcs3*. The combination of FOS stain and the expression pattern of late-expressing genes indicate that the left cluster is the late signature and the right cluster is the early signature. B) Separation of FOS+ early and late signature nuclei (left) or late signature FOS+ and FOS- nuclei (right) for the original (top) and second (bottom) A>A cohorts. Red * = significant association of nucleus state with the principal component. C) Association of the reactivity component with the prediction of reactivity for model i (left) and model ii (right) in the original (top) and second (bottom) A>A cohorts.

REVIEWERS' COMMENTS:

Reviewer #1 (Remarks to the Author):

The authors have sufficiently addressed all questions and concerns raised in the first round.

Reviewer #2 (Remarks to the Author):

In general, the authors have addressed the issues remarked. In particular, they have included some critical control experiments that substantiate their claims.

First of all, authors have now sufficiently validated the use of c-fos marker for neuronal isolation. Data showing similar number of FOS+ neurons obtained by IHC and FACS, as well as the containing Fos-Arc (Rebuttal Figure 1) are enough evidence of this point. Regarding Rebuttal Figure 1, the authors claim that it is included in the final version as Supp Fig 2C. However, the version submitted does not include this panel – is this an error? Also, Supp Fig 2B size has been now excessively reduced, probably by mistake?

The characterization of cfos+ cells in homecage situation – baseline – (Supp Fig 3) is interesting and technically well done. It responds to our request.

Figure 5 has substantially improved and now it helps to understand the strategy used. Also, authors have clarified the reason why they changed the environments A and C between experiments. Their pilot data clearly explains why they have done this. They achieve FAR MORE 'engram' cell labelling with enriched environments. This is fine and understandable but it need so to be made MUCH clearer in the text and in the figures. The reader needs to be able to easily understand that vast majority of the manuscript relies on differential enriched environments as the different contexts, and that only the behavioural figure relies on 'typical' experimental (Fig. 5A). If I was to request one more experiment, I would ask exactly what the cell counting data in Fig. 5F would look like if 'normal' contexts were used instead of enriched environments. I suspect they would be very different and this is important for interpreting the study. If the authors have these data (whether from unpublished or published studies) I would ask that they include this comparison in the manuscript.

Remarkable effort has been put on demonstrating the reproducibility of the model in a new cohort of mice. Authors have tested the probability of reactivity provided by their model.

Minor point: what do "F" and "N" stand for in Supp Fig 6E Legend?

Reviewer #3 (Remarks to the Author):

Authors have responded appropriately to all my concerns in the rebuttal letter.

REVIEWERS' COMMENTS:

Reviewer #1 (Remarks to the Author):

The authors have sufficiently addressed all questions and concerns raised in the first round.

Reviewer #2 (Remarks to the Author):

In general, the authors have addressed the issues remarked. In particular, they have included some critical control experiments that substantiate their claims.

First of all, authors have now sufficiently validated the use of c-fos marker for neuronal isolation. Data showing similar number of FOS+ neurons obtained by IHC and FACS, as well as the containing Fos-Arc (Rebuttal Figure 1) are enough evidence of this point. Regarding Rebuttal Figure 1, the authors claim that it is included in the final version as Supp Fig 2C. However, the version submitted does not include this panel – is this an error? Also, Supp Fig 2B size has been now excessively reduced, probably by mistake?

This was a figure compilation error, we have now corrected Supplemental Figure 5 to include both the IHC and FACS comparison from Rebuttal Figure 1. We have tried to enlarge Supplemental Fig 2b.

The characterization of cfos+ cells in homecage situation – baseline – (Supp Fig 3) is interesting and technically well done. It responds to our request.

Figure 5 has substantially improved and now it helps to understand the strategy used. Also, authors have clarified the reason why they changed the environments A and C between experiments. Their pilot data clearly explains why they have done this. They achieve FAR MORE 'engram' cell labelling with enriched environments. This is fine and understandable but it need so to be made MUCH clearer in the text and in the figures. The reader needs to be able to easily understand that vast majority of the manuscript relies on differential enriched environments as the different contexts, and that only the behavioural figure relies on 'typical' experimental (Fig. 5A). If I was to request one more experiment, I would ask exactly what the cell counting data in Fig. 5F would look like if 'normal' contexts were used instead of enriched environments. I suspect they would be very different and this is important for interpreting the study. If the authors have these data (whether from unpublished or published studies) I would ask that they include this comparison in the manuscript.

We have now included our explanation to the reviewer as part of the Methods section (see: "Animals and treatment" and "Tracking of locomotion during NE exposures"). We have also sought to further clarify this point in the Results section (see "Gene signatures

discriminate reactivation and new activation”) and renamed the behavioral tracking environments to A’ and C’ to distinguish them.

Regarding quantification data similar to Figure 5F but in ‘typical’ environments, we agree that this would be extremely valuable data. However, making the distinction between Newly Activated and Reactivated nuclei relies on having the transcriptome. Quantifying Newly Activated and Reactivated nuclei in different environments would be a non-trivial experiment that would require isolating and performing single-nuclei RNA-Seq on additional batches of cells, taking several months at a minimum to complete.

Remarkable effort has been put on demonstrating the reproducibility of the model in a new cohort of mice. Authors have tested the probability of reactivity provided by their model.

Minor point: what do “F” and “N” stand for in Supp Fig 6E Legend?

These were intended to mean FOS+ and FOS-, respectively. We have now clarified the panel in question.

Reviewer #3 (Remarks to the Author):

Authors have responded appropriately to all my concerns in the rebuttal letter.